# Transcriptional dysregulation by a nucleus-localized aminoacyl-tRNA synthetase associated with Charcot-Marie-Tooth neuropathy

Sven Bervoets [1,12], Na Wei[2,12], Maria-Luise Erfurth[1], Shazie Yusein-Myashkova[1,3], Biljana Ermanoska[1,11], Ligia Mateiu[4], Bob Asselbergh[4], David Blocquel[2], Priyanka Kakad[5], Tyrone Penserga[5], Florian P Thomas[6], Velina Guergueltcheva[7], Ivailo Tournev[8,9], Tanja Godenschwege [5], Albena Jordanova [1,10,13]* & Xiang-Lei Yang [2,13]*

Charcot-Marie-Tooth disease (CMT) is a length-dependent peripheral neuropathy. The aminoacyl-tRNA synthetases constitute the largest protein family implicated in CMT. Aminoacyl-tRNA synthetases are predominantly cytoplasmic, but are also present in the nucleus. Here we show that a nuclear function of tyrosyl-tRNA synthetase (TyrRS) is implicated in a *Drosophila* model of CMT. CMT-causing mutations in TyrRS induce unique conformational changes, which confer capacity for aberrant interactions with transcriptional regulators in the nucleus, leading to transcription factor E2F1 hyperactivation. Using neuronal tissues, we reveal a broad transcriptional regulation network associated with wild-type TyrRS expression, which is disturbed when a CMT-mutant is expressed. Pharmacological inhibition of TyrRS nuclear entry with embelin reduces, whereas genetic nuclear exclusion of mutant TyrRS prevents hallmark phenotypes of CMT in the *Drosophila* model. These data highlight that this translation factor may contribute to transcriptional regulation in neurons, and suggest a therapeutic strategy for CMT.

[1] Molecular Neurogenomics Group, VIB-UAntwerp Center for Molecular Neurology, University of Antwerp, Universiteitsplein 1, 2610 Antwerpen, Belgium. [2] Department of Molecular Medicine, The Scripps Research Institute, 10550 North Torrey Pines Road, La Jolla, CA 92037, USA. [3] Institute of Molecular Biology, Bulgarian Academy of Sciences, Acad. G. Bonchev Str, bl. 21, 1113 Sofia, Bulgaria. [4] VIB-UAntwerp Center for Molecular Neurology, University of Antwerp, Universiteitsplein 1, 2610 Antwerpen, Belgium. [5] Department of Biological Sciences, Florida Atlantic University, 5353 Parkside Drive, Jupiter, FL 33458, USA. [6] Hereditary Neuropathy Foundation Center of Excellence, Department of Neurology, Hackensack Meridian School of Medicine at Seton Hall University, Hackensack University Medical Center, 360 Essex street #303, Hackensack, NJ 07601, USA. [7] Clinic of Neurology, University Hospital Sofiamed, Sofia University St. Kliment Ohridski, bul. G. M. Dimitrov 16, 1797 Sofia, Bulgaria. [8] Clinic of Neurological Diseases, UMBAL Aleksandrovska, Department of Neurology, Medical University-Sofia, ul. Sveti Georgi Sofiyski 1, 1431 Sofia, Bulgaria. [9] Department of Cognitive Science and Psychology, New Bulgarian University, ul. Montevideo 21, 1618 Sofia, Bulgaria. [10] Department of Medical Chemistry and Biochemistry, Medical University-Sofia, ul. Zdrave 2, 1431 Sofia, Bulgaria. [11] Present address: Rosenstiel Basic Medical Sciences Research Center, Department of Biology, Brandeis University, Waltham, MA 02453, USA. [12] These authors contributed equally: Sven Bervoets, Na Wei. [13] These authors jointly supervised this work: Albena Jordanova, Xiang-Lei Yang. *email: albena.jordanova@uantwerpen.vib.be; xlyang@scripps.edu

Charcot-Marie-Tooth (CMT) disease, also known as hereditary motor and sensory neuropathy, is the most common hereditary neuromuscular condition affecting 1 in 2500 individuals[1]. The disease is characterized by weakness and wasting of the distal limb muscles leading to progressive motor impairment, sensory loss, and skeletal deformities. CMT patients typically develop slowly progressive disability early in life, while there is no causal treatment available.

CMT is one of the most heterogeneous Mendelian disorders, with causal mutations in over 80 genes known so far. Notably, CMT gene products are implicated in diverse cellular pathways and their function related to the disease is not always apparent. A prominent example are aminoacyl-tRNA synthetases (aaRSs). With six members involved, they represent the largest family of proteins implicated in the etiology of peripheral neuropathies[2,3]. aaRSs are ubiquitously expressed enzymes catalyzing the charging of tRNAs with their cognate amino acids and are therefore indispensable for viability. The specific sub-population of neurons affected in CMT sharply contrasts with the broad requirement of aaRSs for protein biosynthesis, indicating that the impact of aaRSs in peripheral nerves may not be limited to protein translation.

In addition to their well characterized localization in the cytoplasm, aaRSs are detected in the nucleus of eukaryotic cells. While the early hypothesis was that nuclear aaRSs function in proofreading newly-synthesized tRNAs[4–6], later findings suggest that they are involved in regulating a wide range of biological processes including vascular development, inflammation, and stress responses mainly due to their peculiar abilities to interact with the transcriptional machinery[7–9]. For example, we identified the nuclear localization signal in human tyrosyl-tRNA synthetase (TyrRS or YARS) and demonstrated its presence in the nucleus under oxidative stress to transcriptionally upregulate the expression of DNA damage response genes[8,10]. Overexpression of TyrRS strongly protects against UV-induced DNA damage in zebrafish, whereas restricting TyrRS nuclear entry abolishes this effect. The beneficial effect of nuclear TyrRS is mediated by activating the transcription factor E2F1.

Notably, TyrRS is one of the aaRSs causally linked to CMT disease. Five pathogenic dominant mutations have been reported so far, all of them located in the catalytic domain of the enzyme[11–13]. Three have been extensively characterized both in vitro and in vivo and the results show that a defect in aminoacylation activity is not a shared property (i.e., TyrRS-G41R is enzymatically inactive, while TyrRS-E196K is fully active, and TyrRS-Δ153-156VKQV has partial activity), suggesting that a simple loss of canonical function is not a prerequisite for the disease to occur[14]. Pathogenicity of these three mutations has been recapitulated in transgenic Drosophila models displaying progressive loss of motor abilities, electrophysiological neuronal dysfunction, and terminal axonal degeneration[15]. Flies expressing the enzymatically intact TyrRS-E196K mutant show comparable or, in some aspects, more pronounced features of neurodegeneration than flies expressing the aminoacylation compromised mutants, therefore indicating that a gain of toxic function or interference with a non-enzymatic function of the wild type (WT) protein is likely underlying the disease[15].

In this study, we set out to investigate how this neurotoxic function is generated from a molecular perspective. Because neuronal identity and maintenance are largely controlled by transcriptionally regulated programs[16], we further investigated whether the nuclear localization and function of TyrRS plays any role in the disease mechanism of CMT. We show that CMT-causing mutations in TyrRS induce unique conformational changes, provoking aberrant interactions. These interactions in the nucleus lead to transcription factor E2F1 hyperactivation.

Furthermore, a broad transcriptional regulation network associated with wild-type TyrRS expression in Drosophila is disturbed when a CMT-mutant is expressed. Excluding mutant TyrRS from the nucleus using pharmacological and genetic approaches suppresses the CMT hallmark phenotypes of CMT in the Drosophila model. These data highlight that TyrRS may contribute to transcriptional regulation in neurons, and suggest a therapeutic strategy for CMT.

## Results

**Conformational changes and altered functionalities of TyrRS.** In a previous work, we demonstrated that the three established CMT-causing TyrRS mutants (TyrRS-E196K, TyrRS-G41R, and TyrRS-Δ153-156VKQV) induce a conformational opening and expose a consensus area in the catalytic domain of the enzyme[17] (Fig. 1a, b). In order to link this unique structural change to specific interactional and functional consequences related to CMT, we included two control mutants in this study. An alternative conformational change can be induced by a rationally designed mutation in the anticodon binding domain (Y341A) to expose a different area of the catalytic domain that is responsible for a "cytokine"-like activity of TyrRS[18]. Separately, an established benign polymorphism in the anticodon binding domain (K265N) was included, because it shows no toxicity of the protein in human or when overexpressed in Drosophila and we found that it does not trigger any conformational change[19] (Fig. 1a, b).

Previously, we have found that nuclear TyrRS binds to the scaffolding protein TRIM28[8]. Together, they form a complex with the deacetylase HDAC1 in order to regulate the acetylation level and activity of transcription factors such as E2F1 (Supplementary Fig. 1a). To evaluate how different structural perturbations affect the TyrRS-TRIM28 interaction, we co-expressed the neurotoxic TyrRS mutants (TyrRS-E196K, TyrRS-G41R, and TyrRS-Δ153-156VKQV) or the controls (TyrRS-WT, TyrRS-K265N, and TyrRS-Y341A), together with TRIM28 in HEK293T cells. Our previous studies have suggested that the nuclear localization of TyrRS and its response to oxidative stress are shared among different cell types including HEK293T and motor neuron cells[8]. Immunoprecipitation of TRIM28 revealed co-purification of TyrRS-WT as expected. The CMT-causing mutations enhanced the cellular TyrRS-TRIM28 and TyrRS-HDAC1 interactions (Fig. 1c, d). In contrast, the TyrRS-Y341A mutation resulted in a reduction of the TyrRS-TRIM28/HDAC1 binding, consistent with our previous finding[8], while the benign mutation had no effect on the binding properties, just like the TyrRS-WT protein (Fig. 1c, d). These results demonstrate that the CMT-causing mutants have specific binding properties due to its unique conformational change. We also evaluated the TyrRS-TRIM28 interaction in CMT patient derived peripheral blood mononuclear cells (PBMC) that express the mutant protein in the patients' unique genetic background. The TyrRS-TRIM28 interaction does appear to be stronger in CMT patients' PBMCs endogenously expressing either TyrRS-E196K or TyrRS-G41R mutant alleles (Supplementary Fig. 1b, c). However, the increase did not convey statistical significance given the small number of patient samples available.

Next, we tested if the strengthened TyrRS-TRIM28 interaction would affect the TRIM28/HDAC1 complex to bind its substrate transcription factors (e.g., E2F1) (Supplementary Fig. 1a). Indeed, a reduced TRIM28-E2F1 interaction was concurrent with the enhanced TyrRS-TRIM28 interaction in HEK293T cells expressing each of the three neurotoxic TyrRS alleles (Fig. 1e). As a result, E2F1 was alleviated from deacetylation by HDAC1, and E2F1 acetylation levels were increased (Fig. 1f). Contrarily, the TRIM28-E2F1 interaction was enhanced in cells expressing

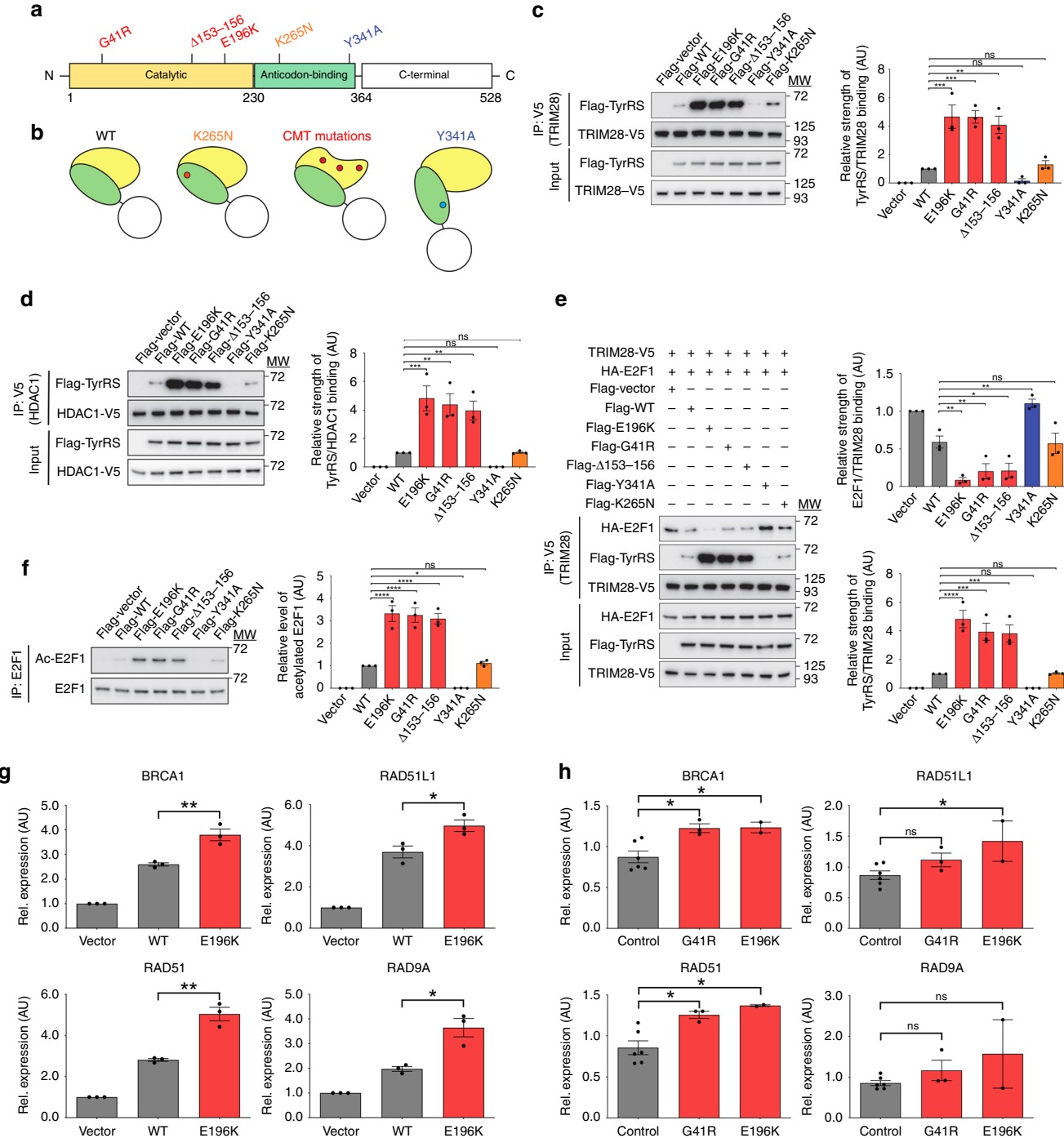

**Fig. 1** TyrRS mutations induce structural changes leading to aberrant transcription regulation. **a** Domain structure of human TyrRS and the location of the different mutations. Red indicates CMT-causing mutations, blue indicates the cytokine-activating Y341A mutation, and orange indicates the benign substitution K265N. **b** Schematic illustration of the conformational changes induced by the different mutations in TyrRS. **c, d** Interaction of TyrRS with TRIM28 (**c**) and HDAC1 (**d**) detected by Co-IP in HEK293T cells expressing different TyrRS proteins. **e** Immunoprecipitation of TRIM28 and the associated binding of E2F1 to the regulatory complex upon expression of different TyrRS alleles. **f** Acetylation levels of E2F1 after IP in HEK293T cells expressing TyrRS. One-way ANOVA with Dunnett Multiple Comparisons test. **g, h** Expression of E2F1 target genes (*BRCA1, RAD51, RAD51L1,* and *RAD9A*) in HEK293T cells expressing different TyrRS alleles (**g**), and patient-derived and control PBMCs (**h**). G41R ($n = 3$ individuals), E196K ($n = 2$ individuals), and control ($n = 5$ individuals). $n = 3$ independent biological replicates, unpaired *t*-test. Bar charts are presented as means ± s.e.m. Source data are provided as a Source Data file

TyrRS-Y341A, leading to a decrease in E2F1 acetylation, while we observed no alterations when the K265N variant was expressed (Fig. 1e, f). These data further demonstrate that the uniquely altered binding properties of CMT mutants have a specific functional impact on their interacting partners.

**CMT-causing mutant TyrRS overactivates E2F1.** Acetylation of E2F1 leads to its activation in human cells[20]. We examined the downstream effect of mutant TyrRS on four E2F1 transcriptional targets sensitive to nuclear TyrRS (*BRCA1, RAD51, RAD51L1, RAD9A*) in mammals[8]. RT-qPCR analysis in HEK293T cells

revealed upregulation of these genes in response to TyrRS-E196K overexpression, which was significantly higher than the response to TyrRS-WT (Fig. 1g). Likewise, we observed a significant increase in E2F1-regulated transcription in CMT patients' PBMCs endogenously expressing mutant TyrRS (Fig. 1h). This upregulation was not due to differences in TyrRS nuclear occupancy, as both HEK293T cells expressing TyrRS-WT or any of the three CMT mutants (G41R, Δ153-156VKQV, E196K), and patient derived PBMCs showed comparable presence of the protein in the nucleus (Supplementary Fig. 2a, b). Thus, via their aberrant interaction properties, CMT-causing mutants behave as hypermorphs regarding their effect on E2F1 activity in mammalian cells.

Next, we explored the relevance of TyrRS-mediated E2F1 overactivation in our Drosophila model for CMT. High expression of TyrRS-E196K in the retina of Drosophila (GMR-Gal4 driver) is toxic and induces a mild rough eye phenotype. In contrast, low expression of TyrRS-WT or TyrRS-E196K shows no retinal disorganization[15] (Fig. 2a–c), but serves as a sensitized background for testing TyrRS-genetic interactors. Expression of the fly orthologue of E2F1 (dE2F1), or its co-factor Dp alone, together with either TyrRS transgene had no effect on the retinal morphology (Fig. 2d–i). However, overexpression of dE2F1 together with Dp induced a rough eye on its own, which remained unaltered in the presence of TyrRS-WT (Fig. 2j, k). In contrast, co-overexpression with TyrRS-E196K aggravated the dE2F1-Dp retinal phenotype as quantified by the reduced eye size on top of the severe rough eye phenotype (Fig. 2l, Supplementary Fig. 3a). The phenotype aggravation is unlikely due to additive toxic effects, considering the lack of any morphological phenotypes upon low level of retinal TyrRS-E196K expression alone (Fig. 2c)[15]. Rather, the result is suggestive of a genetic interaction between TyrRS-E196K and the dE2F1-Dp complex in the fruit fly, consistent with our findings in mammalian cells.

**Inhibition of E2F1 does not rescue TyrRS neurotoxicity in Drosophila.** The overactivation of E2F1 is dependent on its acetylation and may be modulated by activators of known E2F1 deacetylases such as Sirtuin 1 (SIRT1) and HDAC1[21,22]. Indeed, treating HEK293T cells expressing TyrRS-WT or TyrRS-E196K with the HDAC1 activator dexamethasone[23] or SIRT1 activator resveratrol[24] resulted in reduced acetylation levels of E2F1, and the combination of both compounds had a synergistic effect (Fig. 2m). Notably, both drugs affect E2F1 through SIRT1 and HDAC1 specifically, as treatment with either of them in the absence of HDAC1 or SIRT1 (shRNA mediated knockdown) no longer had an effect on E2F1 acetylation (Supplementary Fig. 3b, c).

The success in using dexamethasone and resveratrol to inhibit E2F1 acetylation in cellulo encouraged us to test these compounds in our Drosophila CMT model. Ubiquitous expression (Act5C-Gal4 driver) of the TyrRS-E196K allele induces lethality, as flies have reduced eclosion and pupae formation rates[15,19]. This developmental phenotype is amendable for drug screening and therefore we raised the mutant flies on food containing 100 μM dexamethasone, resveratrol, or both. A small but significant increase in pupae formation was observed upon feeding dexamethasone or the combination of both drugs (Fig. 2n). We then investigated the pharmaceutical effect on the locomotor behavior of the treated flies. Pan-neuronal (nSyb-Gal4 driver) expression of TyrRS-E196K induces severe motor phenotype, which mimics the motor impairment present in the CMT patients, as flies have difficulties climbing a vertical wall[15,25]. No improvement of this climbing deficit was observed upon

administration of resveratrol and dexamethasone (Fig. 2o and Supplementary Fig. 3d). Thus, pharmacological inhibition of E2F1 slightly reduces the overall toxicity of TyrRS-E196K, however it is insufficient to rescue the CMT-related phenotypes in Drosophila. Due to technical difficulty in detecting E2F1 acetylation in Drosophila tissues, we cannot rule out the possibility that the used drugs were not efficient in suppressing E2F1 in the animal. Alternatively or additionally, we speculate that this negative result indicates the neurotoxic effect of the mutant is broader than E2F1 overactivation.

**TyrRS-specific regulatory network altered by a CMT mutant.** Next to TRIM28 and HDAC1, our earlier proteomic studies identified a number of additional nuclear proteins as potential interacting partners of TyrRS-WT in non-neuronal mammalian cells[8]. To assess the full spectrum of downstream effects triggered by TyrRS and its aberrant interactions, we performed expression profiling on whole brains of control (nSyb-Gal4>+), TyrRS-WT, and TyrRS-E196K flies that were aged for 10 days. We opted to study the TyrRS-E196K mutant protein as it does not interfere with the aminoacylation function of the protein and induces the strongest phenotypes in most Drosophila assays[14,15]. In the transcriptome of TyrRS-WT overexpressing flies, 707 genes were differentially expressed compared to the genetic control (nSyb-Gal4>+) (Fig. 3a). These genes were associated with protein folding, cellular stress response, among others, as suggested by the gene ontology (GO) analysis (fgsea[26]), see Supplementary Table 1. Transcription factor (TF) predictions (iRegulon[27]) of both upregulated and downregulated genes identified shared regulatory sequences (E-score >2.5) among the TyrRS-WT regulated genes that are recognized by 38 unique TFs (Supplementary Table 2). Among them was dE2F1, which was in agreement with our previous results, as well as gem, the Drosophila orthologue of TFCP2, a protein identified as an interaction partner of TyrRS in the nucleus of HEK293T cells[8] (Fig. 3b). Thus, the overexpression and the nuclear presence of TyrRS-WT is associated with a network of genes regulated by a multitude of TFs.

Next, we investigated the implications of expressing the TyrRS-E196K mutant on the transcriptome of Drosophila. A total of 830 differentially expressed genes were identified, compared to the genetic control condition (Fig. 3c). GO analysis revealed an enrichment for processes such as neuromuscular synaptic transmission, cellular response to heat, integrin mediated signaling, and protein folding (Supplementary Table 3). A significant overlap was observed between the transcripts differentially expressed in the brains of mutant versus wild-type flies (Fig. 3d). While GO terms overlap between both conditions, there were some distinct differences (Supplementary Fig. 4a and Supplementary Table 4). Of note was the enrichment of neuromuscular synaptic transmission in the E196K condition (Supplementary Fig. 4a).

In addition, weighted gene co-expression network analysis[28] identified one cluster (out of 39 in total) of 415 genes that were co-regulated in a unique manner in the TyrRS-E196K expressing flies only (Fig. 3e, Supplementary Fig. 4b). In this cluster, 227 and 188 genes were upregulated or downregulated, respectively, and the dysregulation of selected candidates was confirmed by RT-qPCR (Supplementary Table 5). iRegulon analysis predicted 53 TFs regulating this gene set (Fig. 3f and Supplementary Table 6). One of the TFs associated with the upregulated genes was dE2F1, for which we already demonstrated experimentally its enhanced activity in a CMT-background and the mechanism behind. Fourteen of the predicted TFs (including dE2F1) were common to both TyrRS-E196K and TyrRS-WT conditions (Supplementary

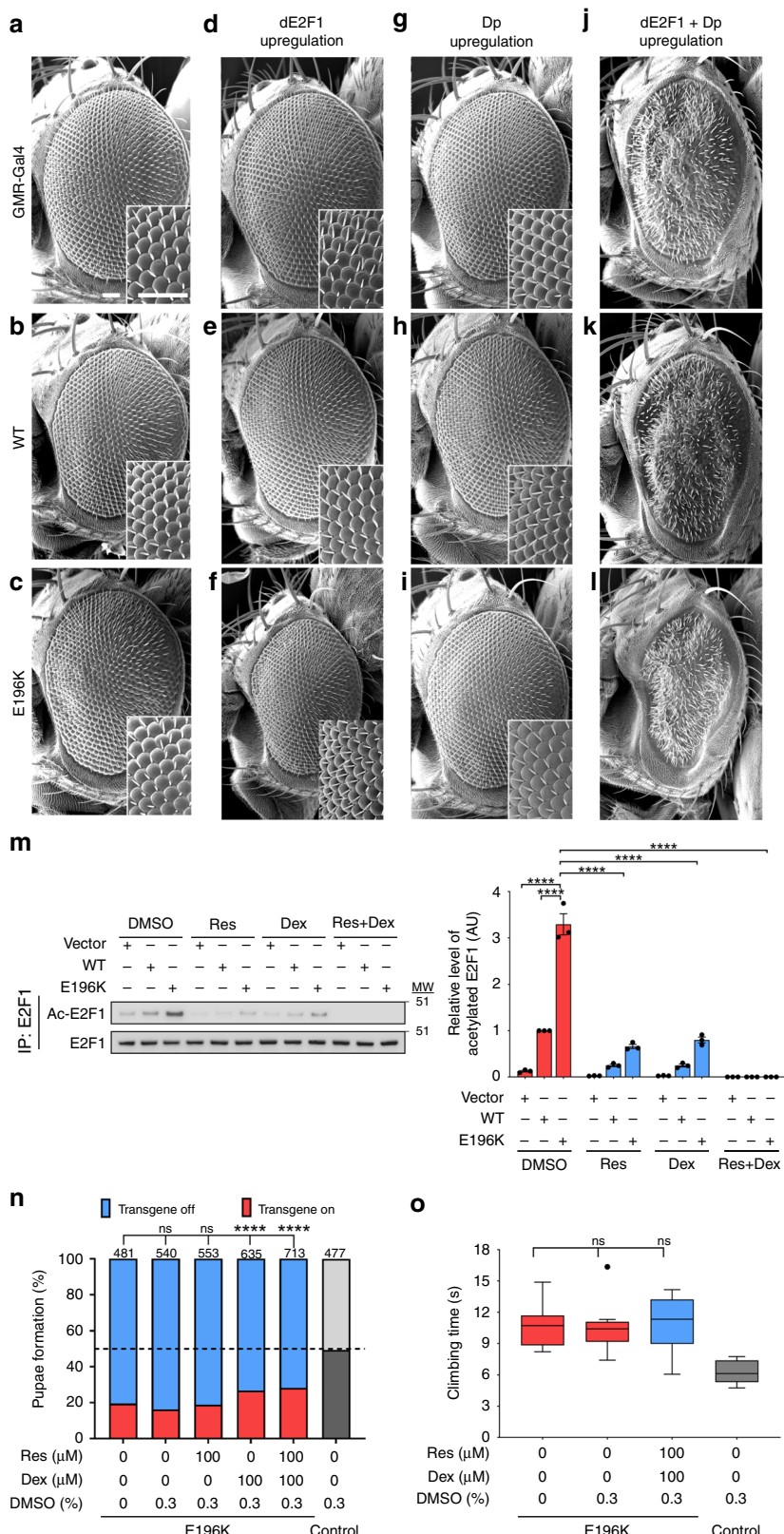

Fig. 4c). This overlap is higher than expected by chance, suggesting that part of the transcript dysregulation observed in the mutant flies is tied to the activity of the WT TyrRS. Notably, 39 additional TFs were linked to the TyrRS-E196K activity only and their function was associated with dendrite morphogenesis, neuronal development and glucose metabolism, among others

(Supplementary Table 7) (DAVID[29,30]). This suggests that the transcriptional impact of the CMT-mutant also has a neomorphic component. Overall, our transcriptome analysis indicates that the CMT-mutant TyrRS affects not only E2F1 but could have a broader impact on the transcriptional landscape in the neurons of the *Drosophila* model.

**Fig. 2** Modulation of E2F1 activity does not rescue disease hallmarks in the TyrRS *Drosophila* model. **a–c** Retinas of control flies (*GMR-Gal4>+*) (**a**), flies expressing TyrRS-WT (**b**), and TyrRS-E196K (**c**) do not display a rough eye phenotype. **d–l** Retinas of flies with a control or TyrRS background that overexpress dE2F1 (**d–f**), Dp (**g–i**), and the combination of both transcription factors (**j–l**). Scale bars, 50 μm. **m** Acetylation of E2F1 after activators of Sirtuin1 (resveratrol (Res)) and HDAC1 (dexamethasone (Dex)) were administered to HEK293T cells expressing different TyrRS alleles. $n = 3$ independent biological replicates, one-way ANOVA with Dunnett Multiple Comparisons test. Bar charts are presented as means ± s.e.m. **n** Pupae formation as an indicator of developmental lethality when *Drosophila* ubiquitously expressing (*Act5C-Gal4*) TyrRS-E196K were grown on resveratrol or dexamethasone. Res and Dex were dissolved in 0.3% DMSO (42 mM). Number of counted flies (*n*) are indicated above the chart, $\chi^2$-test. **o** Climbing performance of flies pan-neuronally expressing (*nSyb-Gal4*) TyrRS-E196K when grown on food containing a combination of resveratrol and dexamethasone. Res and Dex were dissolved in 0.3% DMSO (42 mM). $n \geq 10$ independent groups of flies, one-way ANOVA with Dunnett Multiple Comparisons test. Box plots show the median, 25–75% percentiles, and 1.5 interquartile range. Source data are provided as a Source Data file

**Inhibition of nuclear TyrRS import alleviates neurotoxicity.** In order to demonstrate that the dysregulation in the nucleus by the CMT-mutant is linked to the disease mechanism and knowing that this dysregulation involves multiple genes and associated TFs, we applied a general strategy for alleviating the TyrRS neurotoxicity via inhibiting its nuclear import both genetically and pharmacologically. Nuclear localization of TyrRS is promoted by acetylation, which is mediated by the p300/CBP associated factor (PCAF)[31]. Thus, we attempted to exclude TyrRS from the nucleus using PCAF inhibitors, such as embelin[32]. HEK293T cells showed a dose-dependent reduction in nucleus-localized TyrRS upon titration of embelin (Fig. 4a). These results were confirmed in our *Drosophila* model for CMT, as titration of embelin reduced the nuclear localization of TyrRS (Fig. 4b).

We utilized the previously described developmental lethality assay to investigate the effect of embelin on TyrRS-E196K toxicity in *Drosophila*. A beneficial, dose-dependent drug effect, much stronger than that of dexamethasone and resveratrol, was observed in this experimental paradigm (Fig. 4c). Next, we tested whether embelin would improve the motor performance of flies pan-neuronally (*nSyb-Gal4* driver) expressing TyrRS-E196K. Indeed, at 50 μM and 100 μM concentrations, embelin treated mutant flies climbed significantly faster than untreated animals (Fig. 4d and Supplementary Fig. 5a). Control flies treated with only DMSO or embelin showed no changes in their climbing speed, indicating that embelin did not generally improve climbing behavior of *Drosophila* (Supplementary Fig. 5b). These results show that pharmacological treatment with inhibitors of nuclear entry of TyrRS could partially alleviate the symptoms in the TyrRS-induced CMT model.

The pharmacological restriction of TyrRS nuclear translocation by embelin was not complete (Fig. 4b) and could explain the partial rescue of disease phenotypes in *Drosophila*. As an alternative strategy, we altered the nuclear localization signal (NLS) of TyrRS ($^{242}$KKKLKK$^{247}$) to a less charged sequence ($^{242}$NNKLNK$^{247}$) (hereafter called TyrRS$^{\Delta NLS}$) on both WT and E196K-mutated constructs. We thereby successfully restricted the nuclear import of the ectopically expressed TyrRS proteins in HEK293T cells (Supplementary Fig. 6a). It is worth noting that the NLS is conserved in TyrRS from humans to insects and TyrRS$^{\Delta NLS}$ maintains robust enzymatic activity[10], and could therefore be used to study the TyrRS nuclear function without impacting translation in the cytoplasm. Indeed, while all pan-neuronally (*nSyb-Gal4* driver) expressed human transgenes have comparable expression levels (except for WT TyrRS$^{\Delta NLS}$ with slightly lower expression) (Supplementary Fig. 6b), the nuclear localization of TyrRS$^{\Delta NLS}$ is diminished for both WT TyrRS and the CMT mutant in the *Drosophila* neurons (Fig. 5a). Even for WT TyrRS, the nuclear fraction is very small compared with the cytoplasmic fraction (Supplementary Fig. 6c), therefore the change in nuclear localization would not significantly affect the

amount of TyrRS in the cytoplasm. As such, we have established matching TyrRS and TyrRS$^{\Delta NLS}$ expression systems to genetically study nuclear TyrRS and its link to the CMT pathology.

First, we tested for any general cytotoxicity effect of TyrRS-WT$^{\Delta NLS}$ upon ubiquitous expression, using the developmental lethality assay. *Drosophila* ubiquitously (*Act5C-Gal4* driver) expressing TyrRS-WT or TyrRS-WT$^{\Delta NLS}$ eclosed similarly to the non-transgenic control (Fig. 5b). Remarkably, eclosion rates of animals expressing TyrRS-E196K$^{\Delta NLS}$ were also inconspicuous, suggesting that restriction of mutant protein in the cytoplasm is sufficient to prevent its toxicity (Fig. 5b). Similarly, pan-neuronal (*nSyb-Gal4* driver) expression of TyrRS-E196K$^{\Delta NLS}$ did not cause any locomotor deficits as opposed to the significant climbing impairment observed in the presence of TyrRS-E196K (Fig. 5c and Supplementary Fig. 6d).

We further tested the effect of TyrRS-E196K$^{\Delta NLS}$ expression on the morphology of the *Drosophila* larval neuromuscular junction (NMJ). This synapse is likely the first site of lesion in the dying-back type of neuropathies and has been studied extensively in different CMT fly and mouse models[25,33,34]. To this end, we pan-neuronally expressed (*nSyb-Gal4* driver) different TyrRS transgenes and measured the number of neuronal contact sites (synaptic boutons) and the length of the NMJ. The total length of the neuronal arbor of TyrRS-E196K larvae were smaller and had fewer boutons compared to controls (Fig. 5d-f and Supplementary Fig. 6e, f), showing that NMJ morphology and length are altered upon expression of mutant TyrRS. In contrast, exclusion of TyrRS-E196K from the nucleus resulted in regular NMJ appearance. Excitingly, the transcription fingerprint of TyrRS-WT$^{\Delta NLS}$ and TyrRS-E196K$^{\Delta NLS}$ brains clustered together with the control animals (Fig. 3e and Supplementary Fig. 4b), providing a mechanistic basis for the observed functional rescue. Taken together, our characterization of the TyrRS-E196K$^{\Delta NLS}$ flies at transcriptome, morphological, behavioral, and organismal levels provided strong evidences that the nuclear presence of mutant TyrRS is important for the CMT disease mechanism.

Lastly, to further evaluate the impact of the nuclear mutant TyrRS on CMT neuropathology, we assessed the morphology and synaptic transmission of the *Drosophila* Giant fibers (GFs), containing one of the longest axons in adult flies. We previously used these in the *Drosophila* TyrRS-CMT model and established that the TyrRS-Δ153-156 mutation provokes the strongest GF phenotypes[15]. Using the *R91H05-Gal4* line, we assessed the impact of the cell autonomous expression of different TyrRS proteins in the GFs. Dye injections (Fig. 5g) and electrophysiological recordings (Fig. 5h and Supplementary Fig. 6g) revealed no morphological differences between GF terminals of TyrRS-WT and TyrRS-WT$^{\Delta NLS}$ expressing flies. In contrast, expression of TyrRS-Δ153-156 induced degeneration of the GF terminals (Fig. 5g), which was associated with the reduced ability of the GF synapse to follow repetitive stimuli (Fig. 5h and Supplementary Fig. 6g). Notably, exclusion of the mutant protein from the

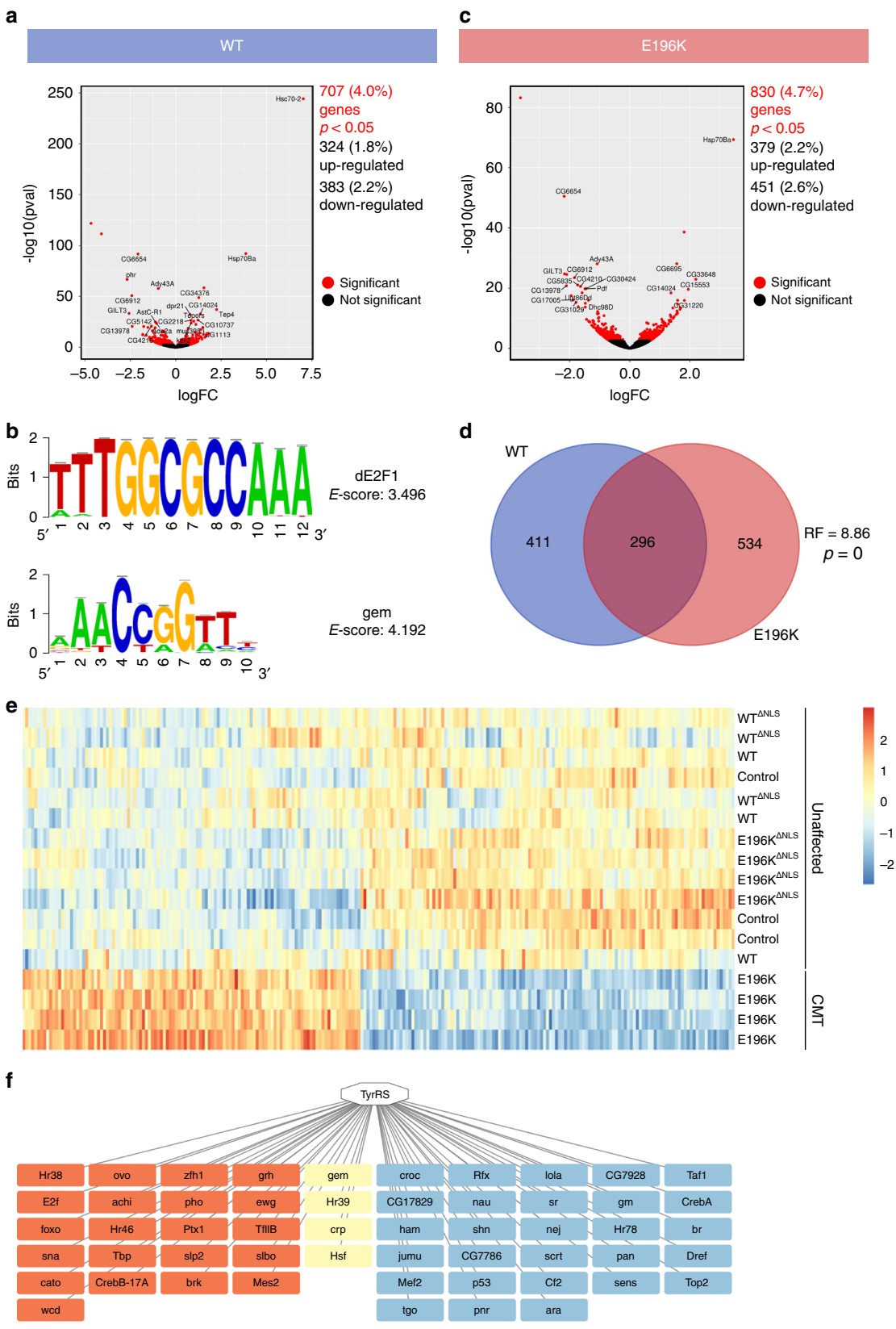

nucleus (TyrRS-Δ153-156$^{\Delta NLS}$) did not provoke these morphological and functional synaptic phenotypes (Fig. 5g, h, and Supplementary Fig. 6g). These results strongly suggest that the GF phenotypes caused by TyrRS-Δ153-156 require its nuclear localization, further supporting that CMT pathology is associated with a dysfunction of mutant TyrRS in the nucleus.

## Discussion

Understanding the pathogenic mechanisms behind the largest protein family linked to CMT, the most common hereditary neuromuscular disorder, imposes a major challenge for both the neuropathy and the tRNA synthetase fields. We and others extensively studied in vitro and in vivo the canonical enzymatic

**Fig. 3** Mutant TyrRS induces broad transcriptional changes in *Drosophila* brain tissues. **a** Volcano plot of log2 fold change versus mean comparing the TyrRS-WT (*nSyb>TyrRS-WT*) with the control condition (*nSyb>+*). **b** Enriched motifs for the dE2F1 and gem transcription factors identified in the upregulated and downregulated genes of the TyrRS-WT condition, respectively. E-score, enrichment score. **c** Volcano plot of log2 fold change versus mean comparing the TyrRS-E196K (*nSyb>TyrRS-E196K*) with the control condition. **d** Venn diagram displaying the overlap between TyrRS-WT and TyrRS-E196K differentially expressed genes. RF, representation factor. *p*, associated probability. **e** Heat map of top 151 differentially expressed genes in the TyrRS-E196K specific 415-gene cluster from brains of *Drosophila* pan-neuronally (*nSyb-Gal4*) expressing the indicated transgenes. High expression is shown in red and low expression is shown in blue (*z*-scores). **f** TFs identified in the TyrRS-E196K specific gene cluster. TFs associated with the downregulated genes are depicted in blue and those associated with the upregulated set are shown in red. The overlapping TFs are indicated in yellow

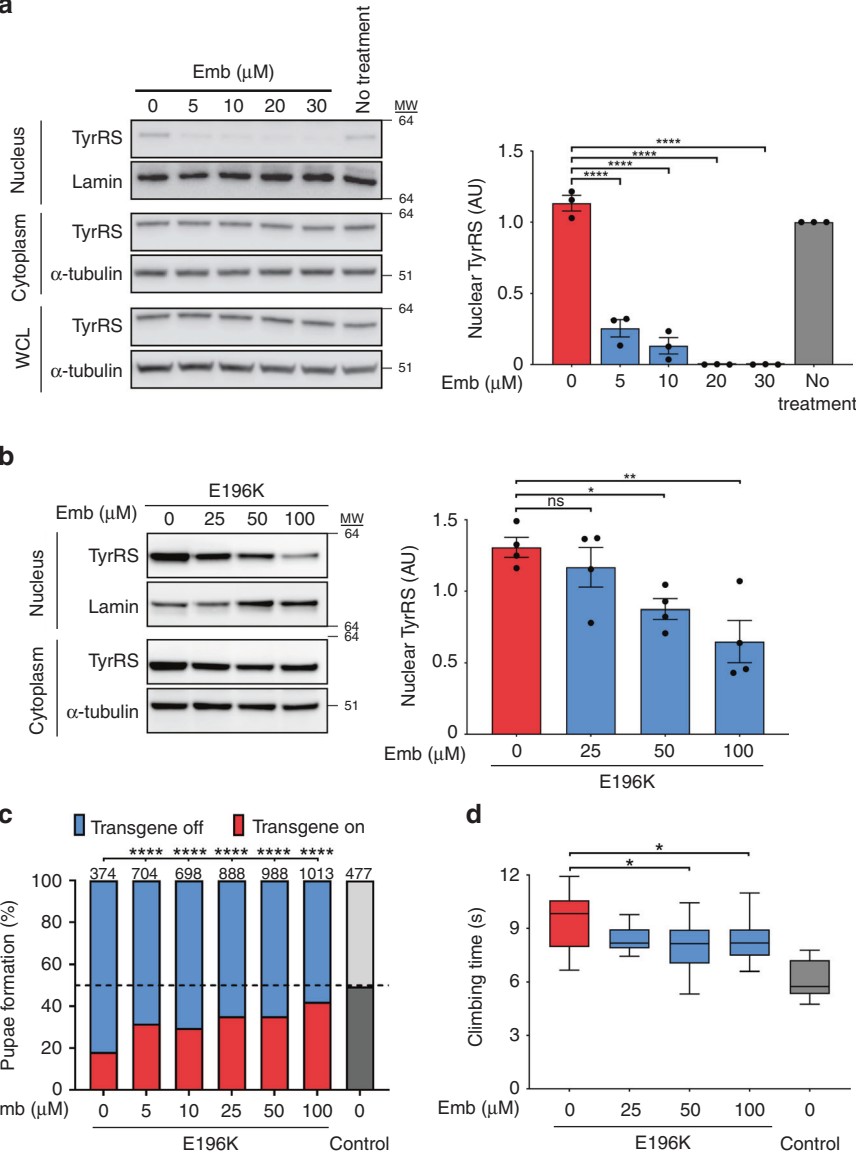

**Fig. 4** Pharmaceutical exclusion of TyrRS from the nucleus partially rescues the toxicity of TyrRS mutants. **a, b** Subcellular localization of TyrRS after treatment with different concentrations of embelin (Emb) in HEK293T cells (14 mM DMSO) (**a**) and in flies pan-neuronally (*nSyb-Gal4*) expressing TyrRS (42 mM DMSO) (**b**). Emb was dissolved in DMSO, which has no effect on TyrRS nuclear localization. α-Tubulin and Lamin served as cytoplasmic and nuclear markers, respectively. *n* ≥ 3 independent biological replicates, one-way ANOVA with Dunnett Multiple Comparisons test. Bar charts presented as means ± s.e.m. **c** Pupae formation of *Drosophila* ubiquitously (*Act5C-Gal4*) expressing TyrRS-E196K after administration of different concentrations of embelin. Emb was dissolved in 0.3% DMSO (42 mM). Percentages indicate ratio of pupae expressing the TyrRS transgene or not. Dashed line, expected 50:50 pupation rate in the control condition (no transgene). Number of counted pupae (n) are indicated above the chart $\chi^2$-test. **d** Climbing performance of flies pan-neuronally (*nSyb-Gal4*) expressing TyrRS-E196K raised on food containing Emb, dissolved in 0.3% DMSO (42 mM). *n* ≥ 10 independent groups of flies. One-way ANOVA with Dunnett Multiple Comparisons test. Box plots show the median, 25–75% percentiles, and 1.5 interquartile range. Source data are provided as a Source Data file

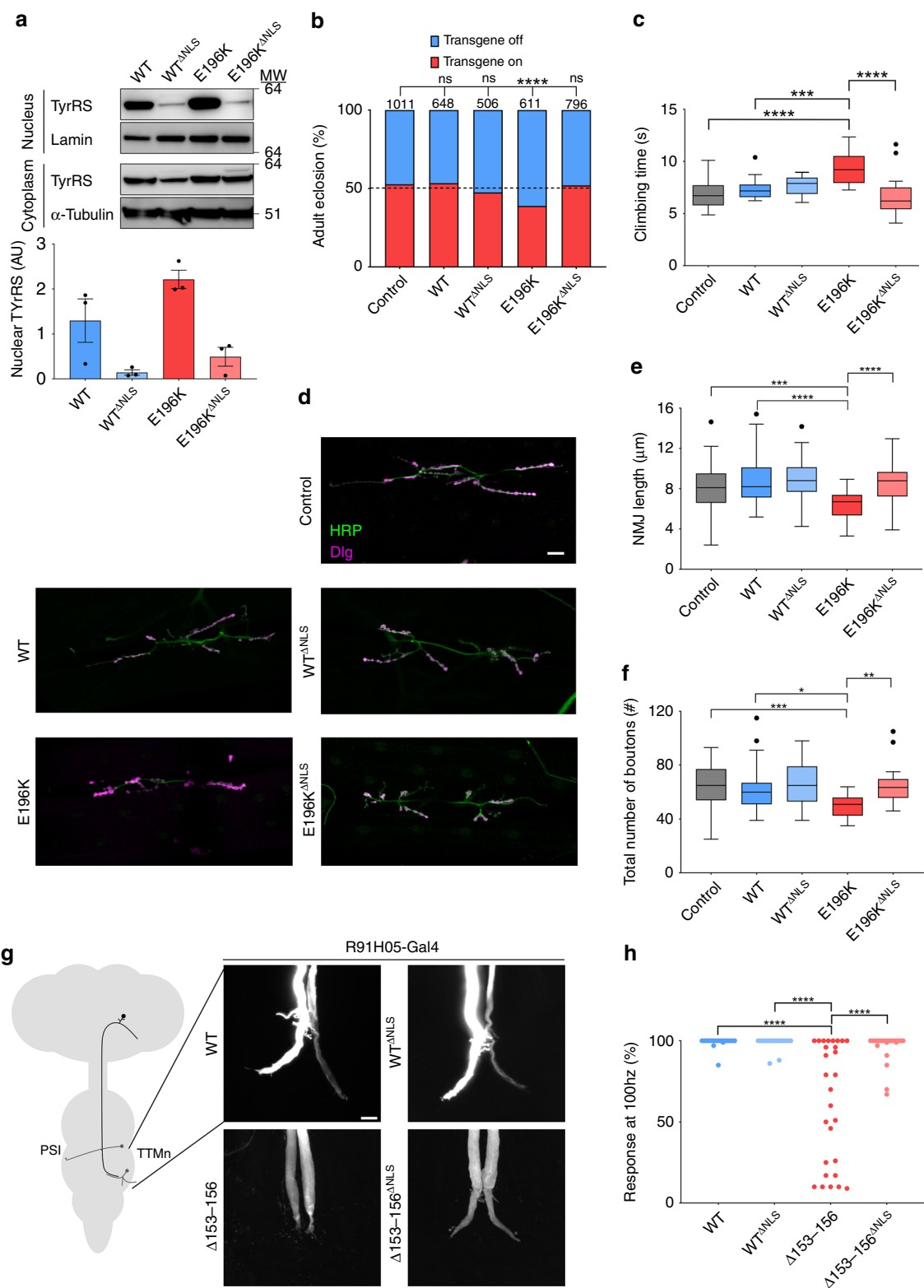

function of aaRSs in relation to CMT and excluded a loss-of-function disease mechanism, at least for two of them (glycyl-tRNA synthetase (GlyRS) and TyrRS)[25]. Although it becomes more and more clear that mutations in these genes cause CMT phenotypes through a toxic gain-of-function mechanism[15,35,36], the molecular and functional aspects of this mechanism remain elusive.

Using a bottom up approach, this study stemmed from our previous understanding of the unique structural changes induced by the CMT-causing mutations in TyrRS[17] (Fig. 1a) and of a role of TyrRS in transcriptional regulation[8]. We searched for functional changes in mutant TyrRS that are relevant to the disease. We took advantage of having a spectrum of TyrRS variants to build a structure-function relationship that specifically links the

**Fig. 5** Genetic nuclear exclusion of mutant TyrRS fully prevents disease phenotypes in *Drosophila* (**a**) Subcellular localization of TyrRS in heads of flies pan-neuronally (*nSyb-Gal4*) expressing different TyrRS alleles. ΔNLS, disrupted NLS motif by mutating $^{242}$KKKLKK$^{247}$ to $^{242}$NNKLNK$^{247}$. α-Tubulin and Lamin served as cytoplasmic and nuclear markers, respectively. $n = 3$ independent biological replicates, bar charts presented as means ± s.e.m. **b** Eclosion rates of flies ubiquitously expressing (*Act5C-Gal4*) no transgene (Control) or different TyrRS-alleles. Dashed line, expected 50:50 eclosion rate in the control condition (no transgene). Number of counted flies (*n*) are indicated above the bars, $\chi^2$-test. **c** Negative geotaxis assay comparing the motor performance of *Drosophila* pan-neuronally (*nSyb-Gal4*) expressing no transgene (Control) or different TyrRS alleles. n ≥ 10 groups of flies. One-way ANOVA with Dunnett Multiple Comparisons test. **d** NMJ's of wandering third instar larvae pan-neuronally (*nSyb-Gal4*) expressing no transgene (Control), TyrRS-WT, TyrRS-WT$^{\Delta NLS}$, TyrRS-E196K or TyrRS-E196K$^{\Delta NLS}$. HRP, neuronal membrane marker; Dlg1, post-synapse marker. Scale bar, 20 μm. **e**, **f** Quantification of total length of the NMJ (**e**) and total number of boutons (**f**). $n \geq 10$ larvae for each genotype. One-way ANOVA with Dunnett Multiple Comparisons test. **g** Dye injections reveal the morphology of the giant fiber terminals in flies expressing (*R91H05-Gal4*) different TyrRS alleles cell autonomously in the giant fibers. Scale bar, 20 μm. **h** The ability of the giant fiber to follow repeated stimuli at 100 Hz upon presynaptic (*R91H05-Gal4*) expression of different TyrRS alleles in the giant fiber. $n \geq 20$ tergotrochanteral muscles (TTMs) of flies aged to 10 days. One-way ANOVA with Dunnett Multiple Comparisons Test. Box plots show the median, 25–75% percentiles, and 1.5 interquartile range. Source data are provided as a Source Data file

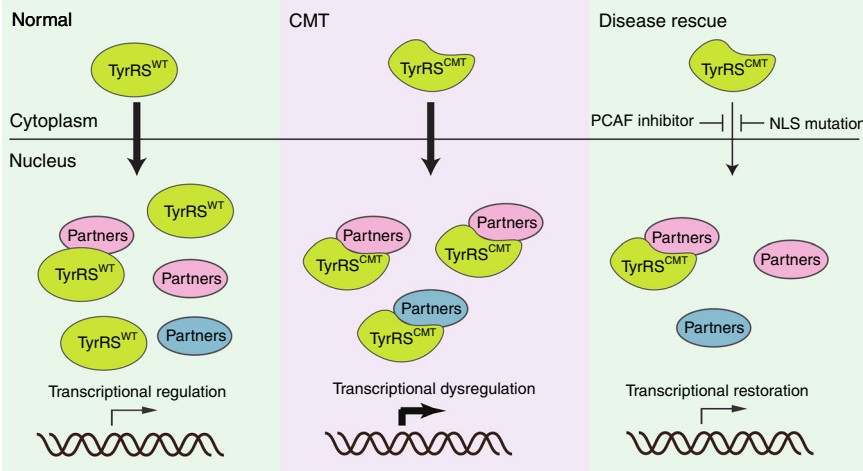

**Fig. 6** Schematic illustration of the proposed TyrRS mode of action. TyrRS enters the nucleus and engages with different interaction partners, including transcription factors. CMT-causing mutations do not affect the nuclear translocation of TyrRS. However, when the mutant protein enters the nucleus, these interactions are enhanced or perturbed and new interactions may be formed, leading to dysregulation of the transcriptional machinery. As demonstrated in a *Drosophila* model, the dysregulation can be alleviated by inhibiting the nuclear entry of TyrRS pharmacologically or genetically

unique conformation of the CMT mutants to functional consequences relevant to the neuropathy. These mutations ranged from a rationally designed substitution, through a naturally occurring polymorphism, to CMT-causing mutations. To this end, we established the mechanism of hyperactivation of E2F1 due to aberrant mutant TyrRS interactions and corroborated our findings in patient biosamples (Fig. 1 and Supplementary Fig. 1a-c). However, the ultimate test for the key involvement of this TF in the pathology would come from a disease rescue in an animal model. Targeting E2F1 reduced the toxicity of TyrRS-E196K but was not sufficient in alleviating the motor deficit of the fly CMT model (Fig. 2n, o). Inspired by our transcriptome analysis, which confirmed the involvement of E2F1 but also revealed a much broader transcriptional dysregulation in mutant flies (Fig. 3), we attempted to target TyrRS nuclear localization and achieved a greater success in reducing toxicity and in alleviating motor deficit (Fig. 4c, d). Remarkably, using a genetic approach to prevent mutant TyrRS from entering the nucleus, we achieved a complete rescue at morphological, behavioral, electrophysiological, and transcriptomic levels altogether (Fig. 5), therefore we unequivocally demonstrated the relevance of the nucleus in TyrRS-linked CMT (Fig. 6). We should point out that although targeting E2F1 alone was not sufficient in rescuing the CMT phenotypes, the molecular mechanism we obtained for E2F1 activation by mutant TyrRS provides a clear proof of concept for how the CMT mutants could affect the function of other transcription factors (Supplementary Fig. 1a).

We note that although the overexpression of WT TyrRS does not cause CMT phenotypes in the fly model[15,25], the overexpression itself is also associated with extensive transcriptional reprograming (Supplementary Table 1), suggesting that this aaRS is a master regulator of gene transcription in the nucleus. Possibly, the WT TyrRS acts through regulating multiple TFs in a coordinated and balanced manner, yet such balance is disrupted by the CMT mutations. The disruption likely has a hypermorphic component as illustrated by E2F1 hyperactivation and a neomorphic component as suggested by the TFs predicted to be regulated by mutant TyrRS alone (Supplementary Fig. 4c). From a structural perspective, the unique conformational change (Fig. 1b) enhances the capacity of CMT-causing TyrRS mutants to interact with transcription regulators, including but should not be limited to TRIM28 and HDAC1, leading to a change in their functional properties and ultimately triggering a broad dysregulation of gene expression.

The demonstration of nuclear involvement in TyrRS-induced CMT could have important implications for other aaRSs associated with CMT, as all aaRSs were described to localize to the nucleus of the cell and at least two other CMT-linked aaRSs (i.e., MetRS and AlaRS) have known functions in the nucleus[37,38]. Interestingly, common genetic modifiers with nuclear localization were found in our previous fly-based screens for CMT-associated mutant GlyRS and TyrRS[39]. This study is facilitated by extensive existing knowledge on the TyrRS system, such as the location of the NLS[10] and the acetylation-mediated TyrRS nuclear import[31].

However, most of this kind of knowledge is lacking for the other CMT-linked aaRS, hindering the evaluation of their possible nuclear involvement at the moment.

Previous works showed that CMT mutations in GlyRS could induce aberrant interactions with ectodomains of membrane receptors (e.g., Nrp1) and intracellular proteins, thereby interfering with proper signaling, as well as trafficking in motor and sensory neurons[35,40–43]. Although we saw a complete rescue of disease-relevant phenotypes in the fly by preventing nuclear entry of the mutant TyrRS, we cannot rule out the involvement of other cellular compartments or additional mechanisms in TyrRS-linked CMT. We did confirm, however, that the ΔNLS mutation introduced to TyrRS did not affect the distribution of the protein (WT or E196K) to the extracellular space (Supplementary Fig. 7a) and that the embelin treatment did not affect the secretion of TyrRS (Supplementary Fig. 7b). Therefore, we excluded the possibility that an altered secretion profile of mutant TyrRS contributed to the genetic and the pharmacological rescue of phenotypes in *Drosophila*. Possibly, the molecules mediating the toxicities outside the nucleus are not as conserved as the mediators inside the nucleus between flies and humans. For example, in contrast to E2F1, there is no direct ortholog of mammalian Nrp1 in *Drosophila*. Developing a mammalian model for TyrRS-linked CMT would facilitate the evaluation of other potential mechanisms in the future.

As illustrated by the studies on GlyRS-linked CMT and by the network of genes and transcription factors identified in our transcriptome analysis for the mutant TyrRS, aaRS-linked CMT is likely a result of the interplay of multiple molecular pathways—some of them active in the WT condition but dysregulated by the aberrant binding properties of the mutant, while others unique to the mutants alone—and therefore the design of a successful therapeutic approach in the future should be tailored to the pleiotropic mutational impact. *Drosophila* can be used to test the therapeutic potential of small molecule compounds in a high-throughput manner[44]. Excitingly, the pharmacological inhibition of TyrRS from entering the nucleus using the PCAF inhibitor embelin mimicked the effect of the genetic restriction and rescued specific disease features in the CMT fly, showing for the first time a successful therapeutic intervention for TyrRS-induced peripheral neuropathy. Although further improvements on drug selection, delivery, and efficacy should be investigated, this work paves the way for developing a viable therapy for CMT patients affected by TyrRS mutations through regulating nuclear trafficking of the translation factor.

## Methods

**Ethical approval.** We have complied with all relevant ethical regulations for animal testing and research. The study was approved by the Ethical committee for biomedical experiments with animals of the University of Antwerp (ECD 2014-02). Informed consent was obtained from all individuals included in the study.

**Drosophila genetics.** The UAS-TyrRS flies were generated and characterized in a previous study[15]. TyrRS$^{\Delta NLS}$ cDNA (WT and with CMT mutations) was subcloned into the pUAST transformation vector and was sequence verified. Transgenic flies were generated using standard procedures. For each construct, multiple transgenic lines were established and expression levels were determined by Western blot. The nSyb-Gal4 driver line was kindly provided by M. Leyssen and B. Dickson[45]. The other Gal4 driver lines and others lines used in the genetic interaction studies were obtained from Bloomington *Drosophila* Stock Center (Supplementary Table 8). *Drosophila* crosses were performed at 25 °C, 12 h light/dark cycle, on standard NutriFly medium (FlyStuff).

The fly line *Act5C-GAL4/Cyo,GFP* was used in the developmental lethality experiments. *Act5C-GAL4/Cyo,RFP-Tb* was used in the pupation experiment. Adult *Drosophila* eclosion ratios were determined by the number of Actin-Gal4>transgene versus the number of Balancer/transgene flies. *Drosophila* pupation was examined by counting the number of Act5C-Gal4>transgene pupae versus the number of Cyo,Tb-RFP>transgene flies.

**Cell culture and conditions.** The HEK293T (human embryonic kidney) cell line was purchased from ATCC (#CRL-3216, Manassas, VA, USA). Cells were grown in DMEM (Gibco) supplemented with 10% heat-inactivated fetal bovine serum (FBS, Gibco). Cultures were maintained at 37 °C in a humidified atmosphere containing 5% $CO_2$.

Upon obtaining written consent, peripheral blood was drawn from CMT patients carrying the E196K or G41R mutations, and control individuals. The sampling procedure was in accordance with the ethical guidelines of the Medical University-Sofia (Bulgaria), the Saint Louis University (USA), and the University of Antwerp (Belgium). Peripheral blood mononuclear cells (PBMC) were isolated on a Ficoll-Paque gradient, transformed with Epstein-Barr virus and incubated at 37 °C for 2 h. After centrifugation, cells were re-suspended in 4 ml RPMI complete medium (Life Technologies) supplemented with 1% phytohaemagglutinin. Cells were seeded on a 24-well plate and incubated at 37 °C, 6% $CO_2$ for a minimum of 3 days. After establishment, cell lines were cultivated in RPMI1640 medium containing 15% fetal calf serum, 1% sodium pyruvate, 1% 200M L-glutamine, and 2% penicillin/streptomycin.

**Co-immunoprecipitation and Western blot analysis.** Lymphoblast and HEK293T cells were lysed with a radioimmunoprecipitation assay (RIPA) buffer containing 25 mM Tris pH 7.6, NaCl, Sodium Deoxycholate, 0.5% NP40, 0.5% Triton, 0.1% SDS, 5% glycerol. Total lysate was incubated with Dynabeads (Life Technologies) or Protein G beads (Life Technologies) and antibodies raised against TyrRS (Abnova), mouse monoclonal V5 (Thermo Fisher Scientific, R960CUS) or mouse monoclonal E2F1 (Abcam, ab4070). Antibodies against IgG1 (Santa Cruz Biotechnology, sc-2025) were used as a negative control. Input and immunoprecipitate were loaded on non-native PAGE gels (Bio-Rad). The following antibodies were used in the different Western blot experiments: mouse monoclonal V5 (1:5000, Thermo Fisher Scientific, R960CUS), mouse monoclonal TyrRS (1:2000, Abnova, H00008565-M02), rabbit polyclonal TyrRS (1:5000, custom-made), rabbit polyclonal TRIM28 (1:1000, Abcam, ab10484), rabbit monoclonal Flag (1:1000, Sigma-Aldrich, F2555), rabbit polyclonal HA (1:1000, Abcam, ab9110), rabbit polyclonal HDAC1 (1:1000, Cell Signaling technology, #2062), rabbit polyclonal SIRT1 (1:1000, Cell Signaling technology, #2310), rabbit polyclonal E2F1 (1:1000, Abcam, ab112580), rabbit polyclonal acetylated-lysine (1:1000, Cell signaling technology, #9441), GAPDH (1:20000, GeneTex, GTX100118), mouse monoclonal α-Tubulin (1:5000, Abcam, ab7291), mouse monoclonal α-Tubulin (1:3000, Cell Signaling Technology, #3873), mouse monoclonal Lamin A/C (1:1000, Cell Signaling technology, #4777), mouse monoclonal Lamin Dm0 (1:500, Developmental Studies Hybridoma Bank, ADL67.10).

The amount of immunoprecipitated, co-immunoprecipitated, and acetylated protein was determined using the ImageJ software. The amount of co-immunoprecipitated or acetylated protein was normalized to the immunoprecipitated signal, resulting in the ratio of both signals.

**Cell fractionation.** Nuclear/cytoplasmic fractionation was performed with either the NE-PER Nuclear and Cytoplasmic Extraction Reagents kit (Thermo Fisher Scientific) following manufacturer's instructions or a similar custom protocol[46]. Briefly, Swelling Buffer (SB) [10 mM Tris-HCl (pH 7.4), 2 mM EDTA, proteinase inhibitor cocktail (added before use)], Plasma Membrane Lysis Buffer (PMLB) [10 mM Tris-HCl (pH 7.1), 2 mM MgCl2, 1% Triton X-100], and Nuclear Extraction Buffer (NEB) [20 mM HEPES (pH 7.6), 300 mM NaCl, 2 mM EDTA, 1 mM 1,4-Dithiothreitol (DTT), 10% glycerol, 1% Triton X-100, protease inhibitor cocktail (added before use)] were prepared. Ice-cold SB was added to the tissue and vortexed at the highest speed for 15 s, followed by incubation on ice for 10 min. Next, ice-cold PMLB (1/20 total volume) was added and vortexed for 5 s and incubated on ice for 1 min. After vortexing for 5 s again, the nuclei were centrifuged at 16,000×g for 5 min at 4 °C. The supernatant (cytoplasm) was transferred to a clean tube and the pellet was washed in PBS, vortexed for 5 s, and the nuclei were centrifuged at 16,000×g for 5 min at 4 °C. The nuclei pellet was then suspended in ice-cold NEB (1/2 of SB volume) and sonicated on ice for 2 × 30 s, with a 30 s pause, using the following settings: cycle 0.5, 10–15% power. Both fractions were then analyzed on Western blot.

**RT-qPCR.** Total RNA was isolated from cell lines following standard Trizol (Qiagen) and chloroform extraction protocol followed by ethanol precipitation. The RNA was then treated with the Turbo DNA-free kit (Ambion) to remove the remaining DNA. RNA was transcribed to single strand cDNA with the iScript cDNA synthesis kit (Bio-Rad). Gene expression levels were determined using the power SYBR green PCR master mix (Applied Biosystems) measured and analyzed with the ViiA7 software (Applied Biosystems) and the qBase + software (Biogazelle). Briefly, five housekeeping genes are used to normalize the data. First, the stability of the housekeeping genes was determined between samples. Next, the most stable housekeeping genes were chosen to normalize the expression of the different genes of interest across all samples. The primers used for the RT-qPCR analysis are described in Supplementary Table 9.

**Scanning electron microscopy.** Adult flies were anesthetised using ether and mounted on aluminum stubs (Electron Microscopy Sciences) without any tissue

processing steps. The eyes were coated using a gold sputter and were imaged using scanning electron microscopy (SEM) with an SEM505 microscope (Philips).

**Drug screen.** *Drosophila* crosses were performed at 25 °C, 12 h light/dark cycle, on Formula 4–24 Instant *Drosophila* medium (Carolina) supplemented with 5% inactivated yeast. DMSO (100%) (Sigma-Aldrich) or embelin (Sigma-Aldrich) were added to the medium at a final DMSO concentration of 0.3% (42 mM). *Drosophila* behavioral assays and pupae formation were performed as described elsewhere.

**Drosophila behavioral assays.** To assay motor performance of 10-days old flies, climbing speed was assessed with the negative geotaxis assay[47]. The semi-automated FlyCrawler device (Peira scientific instruments) was used to test and analyze the different genotypes and conditions. Briefly, 10 female flies with shortened wings were shaken down to the bottom of a cylindrical fly container (49 mm diameter). An infrared camera was used to track individual flies and generate videos. Time needed for the first fly to climb from the start of the ascent at the vertical wall to a mark at a height of 82 mm was measured. For each group of 10 flies, the experiment was done 15 times and the average of the 10 fastest walking speeds was calculated. For each genotype and condition, at least 10 groups of 10 flies were tested.

**Analysis of the Drosophila larval neuromuscular junction.** Morphology of the neuromuscular junction was determined of third instar larvae expressing no transgene (*nSyb-Gal4>+*), or any of the TyrRS alleles. Third instar larvae fillets were dissected in HL3 buffer and fixed in 3.7% PFA in HL3 for 20 min. After permeabilization and blocking of unspecific epitopes (5% Bovine Serum Albumin), staining of the larval NMJ was performed with the following antibodies: horseradish peroxidase (HRP; Jackson ImmunoResearch Laboratories, 1/2000) and Discs large 1 (dlg1; DHSB, 1/50). Images were taken of muscle 6/7 of either segment 3 or 4 using a Carl Zeiss LSM700 laser scanning confocal microscope with ×20 Plan-Apochromat objective (0.8 NA). Maximum intensity projections of image *z*-stacks comprising the entire NMJ were analyzed using the Fiji distribution of ImageJ[48,49]. All boutons were scored using the ImageJ Cell Counter plugin. The length of the neuronal arbor projecting on the muscle was determined using the ImageJ NeuronJ plugin. The arbor was manually traced using the HRP signal.

**Dye injections into the giant fiber.** The giant fiber dye-injection and imaging methods have previously been described in detail[50,51]. The R91H05-Gal4 driver was used to drive the expression of the transgene in the giant fiber[52,53]. Briefly, the GF axons of dissected central nervous systems were injected with tetra-methylrhodamine isothiocyanate-Dextran (Sigma) in the cervical connective using a glass electrode (80–100 MΩ) backfilled with 2 M potassium acetate by passing depolarizing current. The samples were fixed in 4% paraformaldehyde, cleared with VECTASHIELD® Antifade Mounting media and scanned at a resolution of 1024 × 1024 pixels, 2.5 × zoom, and 0.5 μm step size with a Nikon A1 plus confocal microscope using a CFI Plan APO lambda 60×/NA1.4 oil objective. Images were processed using Nikon Elements Advance Research 4.

**Electrophysiological recordings from the giant fiber circuit.** Intracellular recordings from the TTM of adult male and female flies were obtained as previously described[54]. Briefly, the giant fibers (GFs) were activated with 0.03 ms pulses of 30–60 V using two tungsten electrodes inserted into the brain (Grass S44 stimulator, Grass Instruments). Saline-filled glass electrodes were used for recordings from the TTM and a tungsten electrode in the abdomen served as a ground electrode. The recordings were amplified (Getting 5A amplifier, Getting Instruments) and the signals were stored and analyzed using pCLAMP 10.3 software (Molecular Devices). The ability of the GF to TTM pathway to follow high frequency stimulations was assessed with 10 trains of 10 pulses given at 100 Hz with a 1 s interval between the trains. The average following frequencies were calculated as percent responses.

**Total RNA extraction and quality analysis.** Total brain RNA was isolated from adult flies aged 10 days after eclosion, the time point at which mutant flies show locomotor impairment. Total RNA was extracted in quadruplicates using the RNAqueous Micro Kit (Ambion Thermo Fisher Scientific) from dissected adult fly brains with the following genotypes *nSyb-Gal4>+*, *nSyb-Gal4> 2xTyrRS-WT*, *nSyb-Gal4>2xTyrRS-WT$^{\Delta NLS}$*, *nSyb-Gal4>1xTyrRS-E196K*, *nSyb-Gal4>3xTyrRS-E196K$^{\Delta NLS}$*. Following extraction, the total RNA was treated with the Turbo DNA-free kit (Ambion Thermo Fisher Scientific), according to manufacturer's instructions. RNA integrity was assessed using the Eukaryote Total RNA Nano Assay on an Agilent Bioanalyzer 2100 system (Agilent Technologies).

**cDNA library construction and next-generation sequencing.** Twenty cDNA libraries (4 replicates per condition) were constructed from total RNA using poly (A) enrichment of the mRNA (TruSeq Illumina). Paired-end sequencing data of 75 bp read length, generated on NextSeq 500 (Illumina) in 2 batches of 10 samples each, ranged from 74.3 to 139.4 million reads per sample. The cDNA libraries

construction and the RNA sequencing were done at the GENECORE-EMBL Genomic Core Facilities (Germany).

**Bioinformatics analysis.** The raw reads were trimmed and cleaned of adapters with Trimmomatic v0.32. Sequencing quality was assessed with FASTQC software. After trimming and a low reads filtering step, more than 92% of the reads were used for mapping to *Drosophila* reference genome (ENSEMBL, BDGP6, release 84) using Tophat v2.0.12. The average percentage of overall mapping of the reads was 84.2%.

Gene expression levels were measured using the counts generated by HTSeq v0.6.1. Given the deep coverage of the transcriptome, the differential expression analysis was performed on all the genetic features listed in the annotation file resulting in 16237 coding and non-coding genes with non-zero counts in at least one of the samples. More than 95% of the protein coding genes in the genome annotation file had non-zero counts in at least one of the samples. The gene expression counts were normalized for all samples together and the biological conditions were compared pairwise using DESeq2 v1.14.1 by including the batch factor in the design[55]. The PCA and the hierarchical clustering of the regularized log counts revealed 3 outlier samples (*nSyb-Gal4>+*(11), *nSyb-Gal4>TyrRS_WT* (12), *nSyb-Gal4>TyrRS_WT$^{\Delta NLS}$* (18)) which were removed from the final differential expression analysis.

Weighted gene co-expression network analysis was performed (WGCNA) to identify gene clusters that share the same expression patterns across the 5 different biological conditions in 17 samples. The regularized log transformed counts from DESeq2 were used to build an unsigned gene co-expression network. The adjacency matrix computed using a soft threshold of 14 resulted from the scale-free topology calculations. PCA and hierarchical clustering on the rlog transformed gene-level counts within that module were additionally performed. WGCNA constructed 39 gene network modules. The highest eigengene significance (correlation between sample trait and eigengene) with a significant *p*-value was for the magenta module (|0.57|, *p*-value < 0.02). PCA and hierarchical clustering on the rlog transformed gene-level counts within that module were additionally performed. The genes in magenta module showed a clear separation of TyrRS-E196K from the other analyzed conditions.

Gene ontology enrichment was determined using the functional Gene Set Enrichment Analysis (fgsea)[26] using the *Drosophila* gene ontologies derived from the gsea msigdb (v6.2). We included all *Drosophila* transcripts (not only protein coding transcripts) in the fgsea analysis. When fold change data was not available, we determined gene ontology enrichment using the DAVID webtool[29,30]. The iRegulon plugin in Cytoscape was used to detect the transcription factors the targets, and the motifs/tracks associated with the differentially expressed genes.

To calculate and draw custom Venn diagrams the following online tool was used: http://bioinformatics.psb.ugent.be/webtools/Venn/. The representation factor (RF) is the number of overlapping genes divided by the expected number of overlapping genes drawn from two independent groups. A representation factor >1 indicates more overlap than expected of two independent groups, a representation factor <1 indicates less overlap than expected, and a representation factor of 1 indicates that the two groups by the number of genes expected for independent groups of genes. The probability of finding a certain amount of overlapping genes was calculated using the hypergeometric probability formula.

**Secretion assay.** HEK293T cells were cultured to 70% confluency and then transfected with V5-tagged TyrRS or GlyRS constructs. For experiments to test the effect of embelin on secretion, different doses of Embelin or the solvent control (DMSO) were added into the cell culture medium 30 h after transfection. Cell culture medium was collected 36 h after transfection and then concentrated to 1 ml by passing through EMD Millipore Amicon Ultra-15 filter (Millipore Sigma). The secreted TyrRS or GlyRS in the concentrated medium was pulled down by immunoprecipitation by the V5 antibody and then detected by Western blot analysis.

**Statistics.** Measurements were taken from independent samples. We used the GraphPad Prism 7 software, version 7.01, for statistical analyses. Individual tests for each dataset are reported within the figure legends. Reported box plot figures show the median, the box extends from the 25th to the 75th percentiles, and the whiskers depict the 25th and 75th percentile minus and plus 1.5 times the inter-quartile distance, respectively. Reported bar charts show the mean and the error bars as s.e.m. In all figures the *p*-value is reported as *$P < 0.05$, **$P < 0.01$, ***$P < 0.001$, ****$P < 0.0001$. ns, non-significant. The effect sizes were calculated using the PlotsOfDifference webtool[56]. The data points for each experiment were inserted into the web application and the plots were generated using the standard settings. Effect size graphs display the distribution of the differences and the 95% confidence interval is displayed in each graph.

**Reporting summary.** Further information on research design is available in the Nature Research Reporting Summary linked to this article.

## Data availability

All data generated and/or analyzed during this study are included in this published article (and its supplementary files). The source data underlying Figs. 1c–h, 2m–o, 4a–d, 5a–c, e–h, and Supplementary Figs. 1b, c, 2a, b, 3a–d, 5c, 6a–c, 7a, b are provided as a Source Data file. The RNA sequencing data are deposited at the GEO database (GSE125311). All other relevant data are available from the corresponding authors on reasonable request.

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

## Acknowledgements

We would like to thank Ricardo Leitao-Gonçalves for balancing the *Drosophila* lines; Tinne Ooms, and Els De Vriendt for the technical assistance. We acknowledge the service of the Genomics Core Facility of EMBL, Heidelberg, Germany for the RNA sequencing performed. This work was supported in part by the US National Institutes of Health (R01-NS085092 and R01-GM088278 to X.-L.Y.), the US National Institute for Neurological Disease and Stroke (R15NS090043 to T.G.), the Research Fund of the University of Antwerp (TOP-BOF-29069 to A.J.), the Fund for Scientific Research–Flanders (FWO) (G078414N to A.J.) and the AFM-Telethon, France (16197 to A.J.). S.B., B.E., and M.-L.E. obtained fellowships from the Fund for Scientific Research-Flanders. N.W. was supported in part by a fellowship from National Foundation for Cancer Research. S.Y.-M. is the recipient of a DdT2 fellowship from the AFM-Telethon, France. P.K. and T.P. were supported by the Jupiter Life Sciences Initiative of Florida Atlantic University.

## Author contributions

S.B., N.W., A.J., and X.-L.Y. initiated and conceptualized the project. S.B., N.W., M.-L.E., B.E., S.Y.-M, D.B., P.K., T.P., and T.G. performed and analyzed the experiments. F.P.T., V.G., and I.T. aided in patient recruitment and collection of material. L.M. analyzed the RNAseq data and B.A. analyzed the microscopy data. S.B. wrote the original draft, while S.B., N.W., M.-L.E., B.E., S.Y.-M., L.M., B.A., D.B., F.P.T., V.G., I.T., A.J., and X.-L.Y. revised the paper. The work was performed under the supervision of M.-L.E., A.J., and X.-L.Y. A.J. and X.-L.Y. gathered funding.

## Competing interests

The authors declare no competing interests.
