## [Peer Review File · Nature Communications]

Reviewers' comments:

Reviewer #1 (Remarks to the Author):

Here the authors continue their ongoing efforts to define the molecular mechanisms leading to neuronal pathology in Charcot-Marie-Tooth disease (CMT) caused by dominant mutations in aminoacyl-tRNA synthetases, in the present case tyrosyl-tRNA synthetase (TyrRS). The current manuscript expands prior work on the nuclear role of TyrRS in disease pathogenesis. In a 2014 Mol Cell paper the laboratory provided evidence for a nuclear role of TyrRS in protecting from DNA damage through binding to TRIM28, blocking HDAC1 activity, and promotion of an E2F1-dependent DNA damage response. The current manuscript assesses the influence of disease-linked mutations in TyrRS on the same pathway. They find that CMT mutations enhance binding to TRIM28 and promote the expression of DNA damage response genes. The authors then go on to provide evidence that nuclear localization of TyrRS-E196K promotes neurotoxicity in a fly model. These findings are useful follow up to the authors' prior work, but it is not clear that the findings are a significant enough conceptual or technical advance to merit publication in a general interest journal.

Figure 1. It would be helpful to explain in the figure legend what the different colors of the mutations indicate.

Figure 2. These experiments are quite weak.

a-l. Expression of E2F and Dp is very toxic in the fly eye, making interpretation of the putative enhancement of mutant TyrRS challenging.

m-n. The idea is that dexamethasone and resveratrol might alter TyrRS-E196K toxicity by influencing E2F1 acetylation. There is little (n) to no (o) effect on toxicity in flies, but no information is provided on E2F1 acetylation, making the results both negative and hard to interpret.

Figure 4 a,b. Some indication of the reliability of the results should be presented, either multiple blots, quantification, or both.

Figure 4 d. The effects are quite modest.

Figure 5. The effects are quite modest. The data do support the authors' suggestion that nuclear localization of TyrRS-E196K contributes to toxicity, but it is unclear that the phenotype does is robust enough to allow chemical screens, as discussed by the authors.

Reviewer #2 (Remarks to the Author):

Aminoacyl-tRNA synthetases (aaRSs) are the largest protein family implicated in Charcot-Marie-Tooth disease, the most common neuromuscular pathology affecting humans. In this paper, the authors probed the mechanistic basis of CMT-linked TyrRS mutants. A previous study showed that CMT-linked TyrRS mutants undergo a unique conformational change to expose a consensus area near the active site. Previous work also demonstrated a nuclear function for WT TyrRS, which forms a complex with TRIM28/HDAC1 resulting in activation of transcription factor E2F1. In the present study, to link this structural change to biological function, the authors used co-IP and RT-qPCR to evaluate how CMT-causing mutants affect the TRIM28/HDAC1 interaction and E2F1 regulation. These results show that the CMT-linked mutants indeed result in altered binding properties (control mutants did not), establishing a link between the conformational change induced by these mutations and nuclear function. The authors next used an established Drosophila model (ubiquitous expression and pan-neuronal expression of CMT-mutant TyrRS) to evaluate rescue of disease-related neurotoxicity upon drug treatment. Drug-inhibition of E2F1 activity didn't rescue the neurotoxicity, while inhibition of TyrRS nuclear import partially rescued the phenotype. Moreover, using transcriptome analysis the authors identify a TyrRS-specific gene regulatory network, which was altered in the CMT-linked TyrRS mutant. The overall conclusion that a major effect of the CMT mutations is on the nuclear function of TyrRS causing transcriptional

reprogramming, is convincing, although these studies don't rule out additional cytoplasmic effects. The transcriptome analysis provides a good resource for future studies. This study also has broader implications for other CMT-linked aaRSs, as well as for the exciting and rapidly expanding field of aaRS noncanonical functions.

Specific comments:

1. Figure 1b is not very useful. Either the cartoons should be more detailed (e.g., could show different domains by coloring) or this cartoon could be substituted with supplementary Fig.1a, which is helpful to readers for understanding the experimental design in Figure 1.
2. In general, the data presentation and some of the statistical analyses, which are heavily relied on for the major conclusions in this paper, are not as clear or convincing as they could be. In some cases, the conclusions are not supported by the analysis or are stated too strongly.
 - a. All statistical methods and error bars should be clearly defined in the corresponding legends.
 - b. In cases where there are low numbers of replicates, please add data points explicitly to the bar graphs.
 - c. In cases where there are high numbers of replicates, box and whisker plots would be more appropriate.
 - d. Figure 4d: The Embelin rescue effect is not convincing, especially since the 100 μ M concentration is not different from the no-Emb case.
 - e. In Supp Figure 1c, the statistical analysis does not appear to support significance, especially in the case of E196K.
 - f. On page 10, a p value is reported on line 230 but it is not clear how this was calculated and are 6 sig figs really justified?
 - g. The data in Fig 4a do not seem very "dose-dependent" as mentioned in line 245.
3. Is there a specific reason that the authors tested eclosion rate rather than pupae formation in Figure 5b?
4. Modifications to Fig 6: Based on the transcriptome analysis in this work, WT TyrRS and CMT-linked TyrRS mutants result in unique regulatory networks. This result suggests that WT TyrRS is bound with WT-specific binding partners in the nucleus. Is there ever free WT TyrRS in the normal condition or should all TyrRS be bound? Also, the free pink "partners" could be colored blue in the normal and disease-rescue case and CMT-TyrRS would bind to both pink and blue partners to illustrate its altered function.

Minor comment:

1. Mislabeling in supplementary Figure 4: x axis condition: DMSO(%) 0 0.3 0.3

Reviewer #3 (Remarks to the Author):

In their manuscript, Bervoets and coworkers describe the analyses they made to get insights into the molecular mechanism underlying the onset of CMT neuropathy by neomorphic mutations in the human TyrRS gene. They focused on three pathogenic TyrRS mutants: TyrRS-G41R, TyrRS-E196K and TyrRS- Δ 153-156VKQV that change the conformation of regions surrounding TyrRS active site. This study's aims were to verify whether these conformational changes in the TyrRS mutants would change the interactome or functional properties of TyrRS in such a way that the changes could be linked to CMT. For their analyses used 3 different models: HEK293T cells expressing the mutant TyrRS, drosophila flies expressing mutant TyrRS in the retina or pan-neuronally and finally peripheral blood mononuclear cells from patient having these TyrRS CMT mutations. Their results show that the neomorphic TyrRS mutants bind more to TRIM28 thereby decreasing binding of TRIM28 to E2F1 and increased acetylation of E2F1. This increased level of acetylated E2F1 in CMT TyrRS mutant led to retinal disorganization in the fly model. Using their fly model, drugs that either increase deacetylation or inhibit acetylation of E2F1 had a synergistic effect on inhibition of the E2F1 neurotoxic effect mediated by TyrRS CMT mutants. Author then established through comparative transcriptomic between WT TyrRS and CMT mutant TyrRS that there is a specific cluster of transcription factors activated by CMT-causing TyrRS mutants. As a consequence, these TyrRS mutants have a huge impact on the transcriptional repertoire of TyrRS in neurons. This transcriptional change is mediated by the proportion of tyrRS capable of relocating to the nucleus and nuclear relocation of TyrRS depends on its level of acetylation that can be

pharmacologically controlled. Author then checked whether decreasing nuclear import of CMT-causing TyrRS mutants could decrease several disease targets. They compared the size of neuromuscular junctions and the number of neuronal contact sites in flies that pan-neuronally express CMT TyrRS mutants or CMT TyrRS mutants deprived of their nuclear localization signals. These analyses show that that NMJ morphology and size are altered upon expression of CMT mutant TyrRS but that exclusion of CMT TyrRS mutant from the nucleus restores regular NMJ appearance. To conclude, authors provide mechanistic grounds explaining why neomorphic TyrRS mutations provoke CMT phenotypes and show that excluding these mutant TyrRS from the nucleus could be a therapeutic treatment of CMT.

Data collected by authors are solid, their interpretation is overall appropriate and the conclusions reached by the authors is supported by the data; and most importantly significant with respect to CMT disease. I would therefore recommend the publication of this manuscript. However, prior to publication, there are issues that absolutely need to be answered and which are listed below:

Page 6 line 113, Fig 1a & 1b: I find the schematic representation of the impact of the catalytic domain mutations on the overall structure of TyrRS not informative enough: e. g. what is the difference in the structural change of the active site between the 3 CMT mutations and the Y341A mutation? Given that the Y341A mutation lies in the anticodon-binding domain of TyrRS, keeping on fig 1b the color code for the 2 domains (catalytic and anticodon-binding domain) presented in fig 1a would help.

Page 6 line 126: Could authors justify the use of this cell line rather than another one for analyses of mutations that primarily affect motoneurons?

Page 6 line 130: "...while the benign mutation had no effect on the binding properties... Not completely true since when comparing the FlagWT and Flag-K265N (control) lanes, there seem to be a higher binding efficiency with Trim28 for the K265N anticodon mutation than for the WT and less for binding to HDAC1. Quantifications would be needed to be able to conclude that control mutations had no effects.

Page 6 line 131, Fig 1c & 1d: It is not indicated whether these IP experiments have been done once or are representative of several replicates (biological or technical)? Clarify.

Page 7 line 136, Supplementary Figs. 1b, 1c: were levels of TyrRS normalized to GAPDH prior to quantification? Is there a reason why TyrRS- Δ 153-156VKQV PBMC were not tested? Otherwise unpaired t-test OK.

Page 7 line 139: ... reduced TRIM28-E2F1 interaction was concurrent with the enhanced TyrRS-TRIM28 interaction ... Normalized quantifications would be necessary to reach this conclusion. The authors do not say to which variant they compare the CMT mutations to conclude that increased TyrRS-TRIM28 binding correlates with decreased TRIM28-E2F1 interaction: WT or K265N? From the WB observation, I do agree that there is a decreased TRIM28-E2F1 binding for the E196K relative to both the WT and K265N, however, while there is a decreased TRIM28-E2F1 binding for the 2 other CMT mutants relative to K265N, there doesn't seem to be one relative to WT; unless one considers that there is less TRIM28-TyrRS binding for these 2 mutants than for the E196K one. However, there is much less TRIM28-TyrRS binding for the TyrRS- Δ 153-156VKQV mutant than for the G41R one and yet equivalent E2F1 binding. Authors should moderate their statement.

Page 7 line 140 & 142, Fig. 1e & 1f: Again, it is not mentioned whether these IP experiments has been done once or is representative of several replicates

Page 7 line 143-144...no alterations when the K265N variant was expressed... Conclusion should be moderated since the E2F1-TRIM28 binding is significantly enhanced compared to WT TyrRS; and levels of E2F1 acetylation remains higher than for WT TyrRS

Page 7 line 151: ...TyrRS-E196K overexpression... Explain what is the rationale for having chosen this CMT mutant among the 3 possible ones for the directed transcriptomic analyses.

Page 7 line 152, Fig. 1g: From Supp Table 8, one can notice 5 additional pairs of qPCR oligonucleotides that probably correspond to the reference genes used to calculate the relative expressions. However, authors do not indicate how these relative expressions were calculated (neither in the methods section nor in the legend to figure 1). Also, while number of replicates are indicated for PBMCs (fig 1h) they are not for HEK293T cells (fig 1g).

Page 7 line 155: (G41R, Δ 153-156VKQV, E196K): Explain why was the Δ 153-156VKQV mutant not subjected to the transcriptomic analysis.

Page 7 line 156, Supplementary Fig. 1d Nuclear localization was only monitored in the HEK293T cell line (overexpression) and not in PBMCs (endogenous expression), so the upregulation of BRCA1, RAD51, RAD51L1, RAD9A transcription, despite no changes in the level of nuclearly imported TyrRS, only stands for HEK293T cells.

Page 8 line 176, (Fig. 2m) : Interpretation of the data correct, however, no indication of the number of replicates and if this WB is representative of replicates. Also, quantification would greatly improve the clarity of the data. Dexamethasone seem to be less efficient than resveratrol on E2F1 acetylation, is this expected?

Page 8 line 179, Supplementary Figs. 2b, 2c): number of replicates not indicated.

Page 9 line 205, Supplementary Table 1: Number of counts for the GOterms only accounts for 410 genes (57%) from the 700 genes that were found differentially regulated. Could the authors specify why they excluded 43% from the analysis? Alternatively, does it mean that 43% of the regulated transcriptome have less than 3 GOterm counts?

Page 9 line 208: ...dE2F1, which was in agreement with our previous results, as well as gem... Error in the figure legend: Enriched motifs for the dE2F1 and gem transcription factors identified in the up- and down-regulated genes of the TyrRS-WT condition, respectively. SHOULD BE: Enriched motifs for the dE2F1 and gem transcription factors identified in the down- and up-regulated genes of the TyrRS-WT condition, respectively.

Page 9 line 214, Fig. 3c: As a general comment, it would be interesting to have a supp table comparing the WT and E196K GOterm enrichment that allows to have a direct readout on which network of genes sustaining a given pathway is transcriptionally down or up-regulated by the CMT TyrRS mutation.

Page 9 line 216, Supplementary Table 3: Number of counts for the GOterms only accounts for 330 genes (40%) from the 700 genes that were found differentially regulated. Could the authors specify why they excluded 60% from the analysis?

Page 9 line 217, Fig. 3d: Same comment as for fig 3c.

Page 10 line 220, Supplementary Fig. 3a: Color code not explained, areas of clusters not explained.

Page 10 line 243, Fig. 4a: specify number of statistical replicates ? Quantifications would have helped. Authors should indicate the concentrations of DMSO in μ M rather than %.

Page 10 line 244, Fig. 4b: How does 0-100 μ M of embelin for flies compared to 0-30 μ M for HEK293T cells? Also, the control (DMSO) is missing.

Page 11 line 248, Fig. 4c: Explain why pupae formation in absence of 0.3% DMSO for the control or for the E196K has not been measured ?

Page 11 line 250, Fig. 4d: Explain why there is not a dose-dependence amelioration of motor performance upon treatment with embelin, from the fig it seems that there is a 10 % amelioration which plateaus regardless of the embelin concentration used in the experiment while there is a huge difference in nuclear import of TyrRS between 25, 50 and 100 μ M, could the authors comment on this?

Page 11 line 253, Supplementary Fig. 4: Authors are right but differences are not statistically

significant, and there seems to be a mistake since the 2 first histograms are annotated as identical assay conditions (0 μ M embelin 0.3% DMSO).

Page 11 line 264, Fig. 5a: Number of replicates? True, however there is also substantially less TyrRS Δ NLS in the cytoplasm (4th lane), explain why? Also, the lane of the cytoplasmic fraction of E196K variant shows a band of higher molecular weight that is recognized by anti-TyrRS antibodies and not imported inside nuclei. Do the authors know to which form of TyrRS this band corresponds?

Page 12 line 282, Fig. 5d-f: In fig 5e, are the NMJs data between WT E196K significant? This would be needed to sustain the authors' conclusions.

Point-by-point Response to Reviewers' comments:

First of all, we would like to thank all reviewers for their most helpful input and comments. We believe the paper has been strengthened as a result of addressing them. To strengthen our proposed mechanism with regard to neurodegeneration phenotypes, we have performed a new series of *in vivo* analyses through collaboration with the lab of Dr. Tanja Godenschwege at the Florida Atlantic University to assess the morphology and synaptic transmission of the *Drosophila* Giant fibres upon nuclear exclusion of CMT-causing mutant TyrRS. In addition, we have optimized and extended the analysis of the neuromuscular junction morphology in *Drosophila* larval stage and yielded even stronger rescue evidences upon nuclear exclusion of CMT-causing mutant TyrRS. To address concerns raised by the reviewers regarding additional controls and quantifications, use of sufficient replicates, and other clarifications, we have replicated all the experiments with at least three independent biological samples, and rigorously performed quantification and statistical analysis. On the revised manuscript, all major changes we made are highlighted in red; new or updated figures are indicated in green.

Reviewer #1 (Remarks to the Author):

Here the authors continue their ongoing efforts to define the molecular mechanisms leading to neuronal pathology in Charcot-Marie-Tooth disease (CMT) caused by dominant mutations in aminoacyl-tRNA synthetases, in the present case tyrosyl-tRNA synthetase (TyrRS). The current manuscript expands prior work on the nuclear role of TyrRS in disease pathogenesis. In a 2014 Mol Cell paper the laboratory provided evidence for a nuclear role of TyrRS in protecting from DNA damage through binding to TRIM28, blocking HDAC1 activity, and promotion of an E2F1-dependent DNA damage response. The current manuscript assesses the influence of disease-linked mutations in TyrRS on the same pathway. They find that CMT mutations enhance binding to TRIM28 and promote the expression of DNA damage response genes. The authors then go on to provide evidence that nuclear localization of TyrRS-E196K promotes neurotoxicity in a fly model. These findings are useful follow up to the authors' prior work, but it is not clear that the findings are a significant enough conceptual or technical advance to merit publication in a general interest journal.

Figure 1. It would be helpful to explain in the figure legend what the different colors of the mutations indicate.

Thank you for the suggestion. We have now explained the different colors in the figure legend and adapted the figure to make the effect of the mutations on the protein structure clearer.

Figure 2. These experiments are quite weak.

a-l. Expression of E2F and Dp is very toxic in the fly eye, making interpretation of the putative enhancement of mutant TyrRS challenging.

We agree with the Reviewer that due to the toxic effect caused by the E2F1,Dp co-expression on its own in the *Drosophila* retina, the interpretation of the genetic interaction with TyrRS_E196K is challenging and could be regarded as an additive effect rather than a genuine interaction with the mutant

synthetase. To better demonstrate what we think is a genuine genetic interaction between E2F1,Dp and TyrRS-E196K we made the following efforts:

- We tested the same interaction using another publicly available E2F1, Dp fly line where the transgenes are inserted on a different chromosome. Unfortunately, similarly to the results presented in Figs. 2j-l, this second line also induced a strong rough eye phenotype on its own.
- We performed the genetic interaction studies not only at 25°C (the regular temperature for raising fruit flies) but also at 18°C (a lower temperature that does not impair the fly development). Raising flies at lower temperature is a routine manner of reducing the GMR-GAL4 expression and therefore lowering the expression of the E2F1,Dp and TyrRS transgenes so that they have a less toxic effect on eye development. Unfortunately, the co-expression of these two transcription factors at 18°C still resulted in retinal disorganization.
- A common approach in genetic interaction studies when phenotypes *per se* are observed is to quantify the phenotypes observed in flies with different genotypes. To this end, we imaged the fly eyes using light microscopy and measured their size using the ImageJ software. As presented in Supplementary Fig. 3a, co-overexpression of E2F1, Dp and TyrRS wild type does not change the eye surface area. In contrast, co-overexpression of E2F1, Dp and TyrRS-E196K significantly reduces the eye size. Of note, the overexpression of TyrRS-E196K on its own using the same GMR-GAL4 driver does not induce a rough eye phenotype (Fig. 2c). Therefore, the observed aggravation effect in a mutant background is not resulting from an additive effect. We rephrased our statement in the manuscript on page 8 to better describe this particular result. “In contrast, co-overexpression with TyrRS-E196K aggravated the dE2F1-Dp retinal phenotype as quantified by the reduced eye size on top of the severe rough eye phenotype (Fig. 2l, Supplementary Fig. 3a). The phenotype aggravation is unlikely due to additive toxic effects, considering the lack of any morphological phenotypes upon low level of retinal TyrRS-E196K expression alone (Fig. 2c)¹⁵. Rather, the result is suggestive of a genetic interaction between TyrRS-E196K and the dE2F1-Dp complex in the fruit fly, consistent with our findings in mammalian cells.”
- Lastly, we would like to emphasise that the genetic interaction study presented on Figs. 2a-l and the quantification results presented on Supplementary Fig. 3a are only one aspect of demonstrating the interaction between the E2F1 transcriptional complex and mutant TyrRS. We have presented additional evidences for such interaction using alternative approaches throughout the manuscript.
 - o In co-immunoprecipitation experiments in HEK293T cells we show an increased physical interaction between all tested CMT-causing mutations in TyrRS and the E2F1 associated transcription machinery (Figs. 1c-f). Based on the comments of Reviewer 2, we have now included the quantifications of these blots as well.
 - o In our transcriptomics studies in these same HEK293T cells and in patient derived PBMCs endogenously expressing the mutant TyrRS protein we documented upregulation of the E2F1 target genes upon expression of the mutant protein (Figs. 1g, 1h).

All these evidences taken together suggest that mutant TyrRS affects the activity of the E2F1 transcription factor.

m-n. The idea is that dexamethasone and resveratrol might alter TyrRS-E196K toxicity by influencing E2F1 acetylation. There is little (n) to no (o) effect on toxicity in flies, but no information is provided on E2F1 acetylation, making the results both negative and hard to interpret.

We agree that despite the fact that both dexamethasone and resveratrol can lower the acetylation of E2F1 in mammalian cells we were unable to show any phenotypic rescue in our *Drosophila* model. We see at least two explanations for the negative pharmacological outcome. 1) The dexamethasone and resveratrol pharmacokinetics might differ between human cells and *Drosophila* and this could impair the drug efficiency; 2) Lowering the acetylation of E2F1 is insufficient to overcome the mutant TyrRS-related phenotypes observed in *Drosophila*.

In order to address the E2F1 acetylation upon TyrRS overexpression in *Drosophila* we attempted to IP E2F1 from *Drosophila* tissues as we did for the mammalian cells but had no success. Then we co-expressed GFP-E2F1 in flies with different genotypes (no transgene, TyrRS-WT, TyrRS-E196K) using the GMR-Gal4 driver. Immunoprecipitation using a GFP antibody pulled down GFP-E2F1, however, we were unable to detect any acetylated E2F1 on Western blotting (please see figure below). To our knowledge, similar assays on E2F1 acetylation status in *Drosophila* have not been reported in the literature and therefore we could not exclude technical issues.

The second hypothesis however is fully addressed in the manuscript. Based on the subsequent gene expression analysis, we obtained evidence that the dysregulation of E2F1 is not the sole cause of the *Drosophila* phenotypes observed upon mutant TyrRS expression. Broad dysregulation of transcription programs including but not limited to E2F1 was observed. To this end we prevented TyrRS entering the nucleus, which was successful at preventing the disease phenotypes observed in *Drosophila*. This provided further support for our second hypothesis without excluding the first possibility.

We added a new statement in the manuscript: “Due to technical difficulty in detecting E2F1 acetylation in *Drosophila* tissues, we cannot rule out the possibility that the used drugs were not efficient in suppressing E2F1 in the animal. Alternatively or additionally, we speculate that this negative result indicates the neurotoxic effect of the mutant is broader than E2F1 overactivation.”

Figure 4 a,b. Some indication of the reliability of the results should be presented, either multiple blots, quantification, or both.

Following this comment, we repeated the experiment presented on Fig. 4 a,b at least three times, using independent biological samples, and demonstrate the validity of the reduction of TyrRS nuclear localization upon embelin treatment. These quantification data have now been added to the figures.

Figure 4 d. The effects are quite modest.

We agree that the rescue effect of embelin was only partial and we added a sentence in the revised manuscript to acknowledge this partial effect on page 11, which may in part due to the incomplete restriction of TyrRS nuclear localization by embelin. In addition, we have improved our data and data presentation as detailed below.

We have received comments from all three Reviewers about the representation and interpretation of the data distribution in our figures. Therefore, we took special care of improving the representation of the data plots in the revised version of all figures. This includes:

- We added the individual data points to the bar plots where N was low (Figs. 1g, 1h, 5h and Supplementary Figs. 1c).
- If the number of data points was too large to plot, we replaced the bar plots with box plots and whiskers showing: the median, the box (25th to 75th percentiles), and the whiskers based on the Tukey method (1.5 times the inter-quartile difference) (Figs. 2o, 4d, 5c, 5e, 5f and Supplementary Figs. 3a, 5b).
- In case a χ^2 -test was performed, we have added the number of flies that were counted (Figures 2n, 4c, 5b).
- We added an alternative representation of the data distribution, which facilitates the quantitative comparison of the data and displays more clearly the effect sizes of the different experimental conditions. To this end, the mean of the different conditions was resampled with bootstrapping using a tool described in PlotsOfDifference (Goedhart, 2019) – a web app for the quantitative comparison of unpaired data(<https://www.biorxiv.org/content/10.1101/578575v1>) - and shows more clearly the magnitude of the differences between the conditions. By supplying these plots of differences in addition to the classical statistical analyses with p-values, the reader is provided with a direct representation of effect sizes (Supplementary Figs. 3d, 5a, 6d-g)

Concretely on the comment of Fig. 4d, we have added the effect sizes to Supplementary Fig. 5a. This plot shows the distribution of differences after bootstrapping. Furthermore, the black horizontal bar shows the 95% confidence interval associated with this analysis. We have also performed additional measurements and show a significant rescue when mutant flies were treated with 50 μ M and 100 μ M of embelin compared to non-treated flies.

Building on these findings we further validated our nuclear hypothesis using a genetic approach. Using this approach we fully prevented the onset of the disease phenotypes, as demonstrated by the motor performance, NMJ morphology, and giant fibre assays. Thus, our work suggests that there is a room for improvement of the pharmacological interventions as to obtain better effects.

Figure 5. The effects are quite modest. The data do support the authors' suggestion that nuclear localization of TyrRS-E196K contributes to toxicity, but it is unclear that the phenotype does is robust enough to allow chemical screens, as discussed by the authors.

We respectfully disagree that the effect of the genetic rescue is also modest. In fact, in the original manuscript we demonstrated that the exclusion of TyrRS from the nucleus prevents the developmental, motor behavioural, and morphological hallmarks of TyrRS-induced neurodegeneration altogether. In the revised Fig. 5, we now added additional analyses, including assessment of the morphology and synaptic transmission of the *Drosophila* Giant fibres (GFs), which contain one of the longest axons in adult flies and is particularly relevant to the length-dependent neurodegeneration of CMT. Again, exclusion of mutant TyrRS from the nucleus completely rescued the morphological and functional synaptic phenotypes of the GFs. More details on the new analyses we performed, and the robustness of these analyses are explained below.

The phenotypes we tested were selected based on 1) their robustness, 2) relevance to disease, 3) our experience with these assays, and 4) the published examples from others about the application of similar assays in drug screens. In fact, the tested functional read-outs comprise a drug-discovery pipeline of primary and secondary assays for drug discovery in fruit flies developed in Jordanova lab, which was already applied for drug repurposing screens.

- The developmental lethality assay shown in Fig. 5b can be used as a primary screening assay. Strong ubiquitous expression (Act5c-GAL4 or Tubulin-GAL4) of all CMT alleles - but not WT versions of TyrRS - renders dose-dependent developmental lethality occurring in embryonic (10%), larval (80%) or pupal (10%) stages (Storkebaum et al., 2009). This phenotype allows high throughput screening for pharmacological suppressors of mutant TyrRS toxicity, since rescue of lethality is an easy to score phenotype. Similar lethality screens successfully identified drug candidates for another neurodegenerative disorder – the Fragile X syndrome (Chang et al., 2008; Qurashi et al., 2012). Importantly, the lethality is predominantly in the larval stages and therefore is a suitable primary read-out assay to detect the effect of drugs that not only fully rescue the phenotype, but also those that partially suppress it (e.g. an increase of more than 10% pupae upon embelin treatment).
- Neuron-specific expression (*nSyb-Gal4*) of mutant TyrRS renders progressive climbing disability. Importantly, in previous work we have shown that this deficit is not observed in flies expressing TyrRS-WT or the TyrRS-K265N polymorphism. This demonstrates the robustness of the geotaxis assay to evaluate neuronal dysfunction, making it an appropriate validation (orthogonal) assay in a drug screen (Leitao-Goncalves et al., 2012). The negative geotaxis assay has been used before to evaluate the effect of drugs on motor performance (Jia et al., 2008; Qurashi et al., 2012).
- Neuron-specific expression (*nSyb-Gal4*) of mutant TyrRS also affects the morphology of the neuromuscular junction of *Drosophila*, while nuclear exclusion of the mutant synthetase does not induce this effect. Because of the comment of Reviewer 3, we extended the analyses of the NMJ size, which yielded even stronger rescue evidences. In the original manuscript, we calculated the NMJ size as a projected area from the coordinates of the boutons using the smallest bounding rectangle method. While this is a reliable and fast way of measuring NMJ size, it is influenced strongly by the NMJ shape. A more precise extraction of the 'NMJ size' is by measuring the length of the axonal projections innervating the muscle at the NMJ. To do so, we re-evaluated the available images of the NMJ's and measured the total length of the axons in each NMJ by manually tracing the axons based on the HRP staining visualising the neuronal

membrane (the green channel in Fig. 5d). This refined quantification illustrates more clearly the reduced size of the NMJs of the TyrRS-E196K mutant compared to the TyrRS-WT and can be used as an orthogonal assay too (Atkinson et al., 2017; Castells-Nobau et al., 2017). We have also adjusted the Material and Methods section regarding this new analysis.

Lastly, the expert lab of Dr. Tanja Godenschwege at the Florida Atlantic University performed analysis of the giant fibre circuit (GF) in the $\Delta 153-156VKQV$ expressing (*R91H05-Gal4*) adult flies. Together, we have shown previously that expression of mutant - but not WT - TyrRS in the GF and its motor neuron targets results in age-dependent neuronal dysfunction (Ermanoska et al., 2014; Storkebaum et al., 2009). Since the GFS contains the longest axons in *Drosophila*, this progressive phenotype is very relevant for monitoring the length-dependent neurodegeneration, as observed in CMT. Electrophysiological recordings documented that flies expressing CMT-causing TyrRS proteins are unable to follow stimuli one-to-one at high frequencies, as measures of synaptic reliability of their GF circuit. Morphological analyses of the axonal tips shows defasciculation, vacuolization, and constriction. In panels g and h of the revised Fig. 5 we demonstrate that no GF morphological and electrophysiological phenotypes are observed upon TyrRS-WT or TyrRS-WT^{ANLS} expression in the giant fibre. Upon expression of the $\Delta 153-156VKQV$ TyrRS mutant, we observe vacuolisation and constriction of the axon terminal, indicative of length-dependent degeneration. This dying-back axonal phenotype is not observed in the nuclear excluded mutant. Additionally, we analysed in the same flies the electrophysiological properties of the giant fibre upon repetitive stimuli. Again, both WT and WT^{ANLS} conditions show no phenotype, but the $\Delta 153-156VKQV$ flies are unable to sustain an one-to-one response to high frequency stimulations. Exclusion of the del153-156VKQV mutant from the nucleus significantly prevents this phenotype. It is important to emphasize that we have used a different CMT-mutation (TyrRS-153-156delVKQV) in this experiment. The reason is that this mutation provokes the strongest phenotype in the GF assays circuit. This clearly indicates that a second TyrRS mutation leading to the same disease in humans can be prevented in the *Drosophila* model for CMT by excluding its entry in the nucleus.

All the new data has been added to the manuscript and the effect sizes are now available in the supplementary figures.

Reviewer #2 (Remarks to the Author):

Aminoacyl-tRNA synthetases (aaRSs) are the largest protein family implicated in Charcot-Marie-Tooth disease, the most common neuromuscular pathology affecting humans. In this paper, the authors probed the mechanistic basis of CMT-linked TyrRS mutants. A previous study showed that CMT-linked TyrRS mutants undergo a unique conformational change to expose a consensus area near the active site. Previous work also demonstrated a nuclear function for WT TyrRS, which forms a complex with TRIM28/HDAC1 resulting in activation of transcription factor E2F1. In the present study, to link this structural change to biological function, the authors used co-IP and RT-qPCR to evaluate how CMT-causing mutants affect the TRIM28/HDAC1 interaction and E2F1 regulation. These results show that the CMT-linked mutants indeed result in altered binding properties (control mutants did not), establishing a link between the conformational change induced by these mutations and nuclear function. The authors next used an established *Drosophila* model (ubiquitous expression and pan-neuronal expression of CMT-mutant TyrRS) to evaluate rescue of disease-related neurotoxicity upon drug treatment. Drug-inhibition of E2F1 activity didn't rescue the neurotoxicity, while inhibition of TyrRS nuclear import partially rescued the phenotype. Moreover, using transcriptome analysis the authors identify a TyrRS-specific gene regulatory network, which was altered in the CMT-linked TyrRS mutant. The overall conclusion that a major effect of the CMT mutations is on the nuclear function of TyrRS causing transcriptional reprogramming, is convincing, although these studies don't rule out additional cytoplasmic effects. The transcriptome analysis provides a good resource for future studies. This study also has broader implications for other CMT-linked aaRSs, as well as for the exciting and rapidly expanding field of aaRS noncanonical functions.

Specific comments:

1. Figure 1b is not very useful. Either the cartoons should be more detailed (e.g., could show different domains by coloring) or this cartoon could be substituted with supplementary Fig.1a, which is helpful to readers for understanding the experimental design in Figure 1.

We thank all the Reviewers for their suggestions regarding Fig. 1a, 1b. We have now revised Figure 1b with more detailed cartoons including all 3 domains of TyrRS to better illustrate the mutation-induced conformational changes.

2. In general, the data presentation and some of the statistical analyses, which are heavily relied on for the major conclusions in this paper, are not as clear or convincing as they could be. In some cases, the conclusions are not supported by the analysis or are stated too strongly.

a. All statistical methods and error bars should be clearly defined in the corresponding legends.

b. In cases where there are low numbers of replicates, please add data points explicitly to the bar graphs.

c. In cases where there are high numbers of replicates, box and whisker plots would be more appropriate.

We thank the Reviewer for suggesting ways to improve the presentation of our data as to support our statements. We made considerable efforts to re-analyse or represent our findings in a better way. All

figures and their legends were revised and we add additional figures graphically demonstrating the effect sizes.

For example:

- We added the individual data points to the bar plots where N was low (Figs. 1g, 1h, 5h and Supplementary Figs. 1c).
- If the number of data points was too large to plot, we replaced the bar plots with box plots and whiskers showing: the median, the box (25th to 75th percentiles), and the whiskers based on the Tukey method (1.5 times the inter-quartile difference) (Figs. 2o, 4d, 5c, 5e, 5f and Supplementary Figs. 3a, 5b).
- In case a χ^2 -test was performed, we have added the number of flies that were counted (Figures 2n, 4c, 5b).
- We added an alternative representation of the data distribution, which facilitates the quantitative comparison of the data and displays more clearly the effect sizes of the different experimental conditions. To this end, the mean of the different conditions was resampled with bootstrapping using a tool described in PlotsOfDifference (Goedhart, 2019) – a web app for the quantitative comparison of unpaired data(<https://www.biorxiv.org/content/10.1101/578575v1>) - and shows more clearly the magnitude of the differences between the conditions. By supplying these plots of differences in addition to the classical statistical analyses with p-values, the reader is provided with a direct representation of effect sizes (Supplementary Figs. 3d, 5a, 6d-g)

Please see also our elaborate answer to a similar comment of Reviewer 1.

d. Figure 4d: The Embelin rescue effect is not convincing, especially since the 100 μ M concentration is not different from the no-Emb case.

Please see also our answers to a similar comment by Reviewer 1. In the revised manuscript we have added new measurements of the fly climbing behaviour as well as the associated effect sizes. A significant rescue in motor performance is observed upon treating mutant expressing flies with 50 μ M and 100 μ M of embelin (See revised Fig. 4d). Nevertheless, we agree this is a modest rescue. We noticed that in *Drosophila* the nuclear exclusion upon treatment with embelin was not complete (quantification in Fig. 4b), thus provoking our efforts to test the nuclear exclusion of TyrRS with a different approach. Genetic exclusion of mutant TyrRS had a much stronger beneficial effect on the observed phenotypes in the *Drosophila* model for CMT, indicating that further improvements to the pharmaceutical strategy are needed.

e. In Supp Figure 1c, the statistical analysis does not appear to support significance, especially in the case of E196K.

We agree. Supplementary Fig. 1c shows the average ratio of three independent experiments of pulled down TRIM28 per TyrRS, hence control = 3 individuals, G41R=3 individuals, and E196K= 2 individuals. We chose to analyse this experiment based on biological replicates (different family members). There are several possibilities to why our analysis does not support significance. 1) We are limited to the number of CMT individuals available, therefore, our sample size is low. This affects the power of our statistical analysis. 2) The interaction observed in patients' PBMCs is not as pronounced as in the HEK293T cells. The reason for this is that in HEK293T cells we overexpressed TyrRS and TRIM28, thus

making an interaction between both proteins more obvious. The power of the experiment in the PBMCs resides within the fact that these cells express the proteins at endogenous levels and in a patient specific genetic background. We are able to detect the interaction between TyrRS and TRIM28 in this setup, albeit not significant. 3) Building on the previous point, Western blot is a semi-quantitative analysis, which is unable to pick subtle differences.

We have adjusted the text in the new version of the manuscript: “The TyrRS-TRIM28 interaction does appear to be stronger in CMT patients’ PBMCs endogenously expressing either TyrRS-E196K or TyrRS-G41R mutant alleles (Supplementary Figs. 1b, 1c). However, the increase did not convey statistical significance given the small number of patient samples available”. Furthermore, we now show the individual data points in the graph, for better data interpretation, and the p-values for the different comparisons based on the unpaired t-test.

f. On page 10, a p value is reported on line 230 but it is not clear how this was calculated and are 6 sig figs really justified?

The value reported depicts the probability (P) of finding a certain amount of overlapping genes, which was calculated using the hypergeometric probability formula. To avoid confusions, we have removed the value in the text, as nowhere else we mention such values. The probability (P) is still included in the relevant figures.

g. The data in Fig 4a do not seem very “dose-dependent” as mentioned in line 245.

We have repeated the experiments presented in Fig. 4a three times, using independent biological samples, which allow us to quantify the result. As show in the quantification, there is indeed a dose-dependent effect of embelin in restricting TyrRS nuclear localization.

3. Is there a specific reason that the authors tested eclosion rate rather than pupae formation in Figure 5b?

Strong ubiquitous expression (*Act5C-GAL4* or *Tubulin-GAL4*) of all CMT alleles - but not WT versions - of TyrRS renders dose-dependent developmental lethality occurring in embryonic (10%), larval (80%) or pupal (10%) stages (Storkebaum et al., 2009). When the TyrRS-E196K protein is expressed, this lethality is full and no adult flies eclose (in case of strong ubiquitous expression) or partial, i.e. there is 10% adult fly eclosion (weak ubiquitous expression) (Storkebaum et al., 2009). We chose to test the effect of nuclear exclusion of TyrRS-E196K on adult eclosion rate because in this way we monitor all developmental stages of *Drosophila* and thus our read out is as thorough/robust as possible. Please see also our answer to a related question of Reviewer 1.

4. Modifications to Fig 6: Based on the transcriptome analysis in this work, WT TyrRS and CMT-linked TyrRS mutants result in unique regulatory networks. This result suggests that WT TyrRS is bound with WT-specific binding partners in the nucleus. Is there ever free WT TyrRS in the normal condition or should all TyrRS be bound? Also, the free pink “partners” could be colored blue in the normal and disease-rescue case and CMT-TyrRS would bind to both pink and blue partners to illustrate its altered function.

We appreciate the suggestion, Fig. 6 has now been updated.

Minor comment:

1. Mislabeling in supplementary Figure 4: x axis condition: DMSO(%) 0 0.3 0.3

The mislabeling has been corrected.

Reviewer #3 (Remarks to the Author):

In their manuscript, Bervoets and coworkers describe the analyses they made to get insights into the molecular mechanism underlying the onset of CMT neuropathy by neomorphic mutations in the human TyrRS gene. They focused on three pathogenic TyrRS mutants: TyrRS-G41R, TyrRS-E196K and TyrRS- Δ 153-156VKQV that change the conformation of regions surrounding TyrRS active site. This study's aims were to verify whether these conformational changes in the TyrRS mutants would change the interactome or functional properties of TyrRS in such a way that the changes could be linked to CMT. For their analyses used 3 different models: HEK293T cells expressing the mutant TyrRS, drosophila flies expressing mutant TyrRS in the retina or pan-neuronally and finally peripheral blood mononuclear cells from patient having these TyrRS CMT mutations.

Their results show that the neomorphic TyrRS mutants bind more to TRIM28 thereby decreasing binding of TRIM28 to E2F1 and increased acetylation of E2F1. This increased level of acetylated E2F1 in CMT TyrRS mutant led to retinal disorganization in the fly model. Using their fly model, drugs that either increase deacetylation or inhibit acetylation of E2F1 had a synergistic effect on inhibition of the E2F1 neurotoxic effect mediated by TyrRS CMT mutants. Author then established through comparative transcriptomic between WT TyrRS and CMT mutant TyrRS that there is a specific cluster of transcription factors activated by CMT-causing TyrRS mutants. As a consequence, these TyrRS mutants have a huge impact on the transcriptional repertoire of TyrRS in neurons. This transcriptional change is mediated by the proportion of tyrRS capable of relocating to the nucleus and nuclear relocation of TyrRS depends on its level of acetylation that can be pharmacologically controlled. Author then checked whether decreasing nuclear import of CMT-causing TyrRS mutants could decrease several disease targets. They compared the size of neuromuscular junctions and the number of neuronal contact sites in flies that pan-neuronally express CMT TyrRS mutants or CMT TyrRS mutants deprived of their nuclear localization signals. These analyses show that that NMJ morphology and size are altered upon expression of CMT mutant TyrRS but that exclusion of CMT TyrRS mutant from the nucleus restores regular NMJ appearance. To conclude, authors provide mechanistic grounds explaining why neomorphic TyrRS mutations provoke CMT phenotypes and show that excluding these mutant TyrRS from the nucleus could be a therapeutic treatment of CMT.

Data collected by authors are solid, their interpretation is overall appropriate and the conclusions reached by the authors is supported by the data; and most importantly significant with respect to CMT disease. I would therefore recommend the publication of this manuscript. However, prior to publication, there are issues that absolutely need to be answered and which are listed below:

Page 6 line 113, Fig 1a & 1b: I find the schematic representation of the impact of the catalytic domain mutations on the overall structure of TyrRS not informative enough: e. g. what is the difference in the structural change of the active site between the 3 CMT mutations and the Y341A mutation? Given that

the Y341A mutation lies in the anticodon-binding domain of TyrRS, keeping on fig 1b the color code for the 2 domains (catalytic and anticodon-binding domain) presented in fig 1a would help.

Again, we thank all Reviewers for making this suggestion. We have now revised Fig. 1b with more detailed cartoons with all 3 domains of TyrRS included to better illustrate the mutation-induced conformational changes.

Page 6 line 126: Could authors justify the use of this cell line rather than another one for analyses of mutations that primarily affect motoneurons?

We used HEK293T cells here for its transfection efficiency to carry out biochemical analysis with mostly ectopically expressed proteins. Our previous studies have suggested that the nuclear localization of TyrRS and its response to oxidative stress are shared among different cell types including HEK293T cells and motor neuron cells (e.g., MN-1) (Wei et al., 2014). We now add this statement/justification in the manuscript on page 6.

Page 6 line 130: "...while the benign mutation had no effect on the binding properties... Not completely true since when comparing the FlagWT and Flag-K265N (control) lanes, there seem to be a higher binding efficiency with Trim28 for the K265N anticodon mutation than for the WT and less for binding to HDAC1. Quantifications would be needed to be able to conclude that control mutations had no effects.

We have repeated the experiments presented in Figs. 1c and 1d three times, with independent biological samples, which allows us to quantify the result. These quantifications are now added as bar charts in Figs. 1c and 1d. As shown in the quantification, there is no significant difference between TyrRS-WT and TyrRS-K265N in regard to their binding with TRIM28/HDAC1.

Page 6 line 131, Fig 1c & 1d: It is not indicated whether these IP experiments have been done once or are representative of several replicates (biological or technical)? Clarify.

As described above, the IP experiments in Figs. 1c and 1d has now been repeated three times, with new cells and transfections each time (biological replications).

Page 7 line 136, Supplementary Figs. 1b, 1c: were levels of TyrRS normalized to GAPDH prior to quantification? Is there a reason why TyrRS- Δ 153-156VKQV PBMC were not tested? Otherwise unpaired t-test OK.

The amount of co-pulled down TRIM28 was normalized to the amount of pulled down TyrRS, without any further GAPDH normalisation. The amount of antibody used for the pull down in every experiment was the same and the amount of co-immunoprecipitated GAPDH was comparable between the different conditions. This information was now added to the Material and Methods section on page 21 of the revised manuscript.

In contrasts to the TyrRS-E196K and TyrRS-G41R mutations which were found in large multigeneration pedigrees with many affected family members, the TyrRS- Δ 153-156VKQV mutation is a de novo deletion identified in a single CMT patient in the world. Unfortunately, this unique patient never donated PBMCs and therefore we could not perform any test in biomaterials from this individual.

Page 7 line 139: ... reduced TRIM28-E2F1 interaction was concurrent with the enhanced TyrRS-TRIM28

interaction ... Normalized quantifications would be necessary to reach this conclusion. The authors do not say to which variant they compare the CMT mutations to conclude that increased TyrRS-TRIM28 binding correlates with decreased TRIM28-E2F1 interaction: WT or K265N? From the WB observation, I do agree that there is a decreased TRIM28-E2F1 binding for the E196K relative to both the WT and K265N, however, while there is a decreased TRIM28-E2F1 binding for the 2 other CMT mutants relative to K265N, there doesn't seem to be one relative to WT; unless one considers that there is less TRIM28-TyrRS binding for these 2 mutants than for the E196K one. However, there is much less TRIM28-TyrRS binding for the TyrRS- Δ 153-156VKQV mutant than for the G41R one and yet equivalent E2F1 binding. Authors should moderate their statement.

We have repeated the experiment presented in Fig. 1e three times, with independent biological samples, to allow us to quantify the result. All variants were compared to the TyrRS-WT condition. The quantification, which is now included in Fig. 1e, shows TyrRS-E196K is indeed significantly different compared to the TyrRS-WT condition. Such an observation is now also obvious for the other two remaining CMT mutations. Taken together, the new quantification supports our original statement in the manuscript.

Page 7 line 140 & 142, Fig. 1e & 1f: Again, it is not mentioned whether these IP experiments has been done once or is representative of several replicates

Both experiments in Fig. 1e and 1f have now been repeated three times, using new cells, and new transfections each time. One representative figure of each experiment is shown in Figs. 1e and 1f. We also added the quantification of the Western blots as bar charts and the accompanying statistical analyses.

Page 7 line 143-144...no alterations when the K265N variant was expressed... Conclusion should be moderated since the E2F1-TRIM28 binding is significantly enhanced compared to WT TyrRS; and levels of E2F1 acetylation remains higher than for WT TyrRS

We have included the quantification of three independent Western blots in the figure, where the intensity of the acetylated-E2F1 band is measured. The statistical analyses regarding this quantification is also included. Based on the quantification, there is no significant difference in E2F1 acetylation level between TyrRS-WT and TyrRS-K265N cells. We appreciate these comments, which prompt us to clarify on these points.

Page 7 line 151: ...TyrRS-E196K overexpression... Explain what is the rationale for having chosen this CMT mutant among the 3 possible ones for the directed transcriptomic analyses.

The primary reason is that we and others have shown both in vitro and in vivo that this mutation does not affect the aminoacylation function of TyrRS (Froelich and First, 2011; Storkebaum et al., 2009). Hence, any dominant-negative toxic effect resulting from loss of aminoacylation is avoided. In addition, expression of this mutation induces the strongest phenotypes in most of the assays in *Drosophila*. We have now added a sentence in page 9 to explain the selection of the TyrRS-E196K mutation.

Page 7 line 152, Fig. 1g: From Supp Table 8, one can notice 5 additional pairs of qPCR oligonucleotides that probably correspond to the reference genes used to calculate the relative expressions. However,

author do not indicate how these relative expressions were calculated (neither in the methods section nor in the legend to figure 1). Also, while number of replicates are indicated for PBMCs (fig 1h) they are not for HEK293T cells (fig 1g).

It is a custom practice in the Jordanova lab to use 5 reference genes to normalise the amount of starting material. The relative expression of the different genes was established using the qbase+ software from Biogazelle. This analysis is performed in two consecutive steps: 1) the stability of the different reference genes is estimated across the different samples. 2) the most stable reference genes are used to calculate the relative expression of the gene of interest across different samples. This information was added to the Material and Methods section of the manuscript.

We have now also added a statement in the figure legend regarding the number of replicates of the HEK293T cells.

Page 7 line 155: (G41R, Δ 153-156VKQV, E196K): Explain why was the Δ 153-156VKQV mutant not subjected to the transcriptomic analysis.

We do not have biomaterials from this unique patient. Please see also our answer on a similar question above.

Page 7 line 156, Supplementary Fig. 1d Nuclear localization was only monitored in the HEK293T cell line (overexpression) and not in PBMCs (endogenous expression), so the upregulation of BRCA1, RAD51, RAD51L1, RAD9A transcription, despite no changes in the level of nuclearly imported TyrRS, only stands for HEK193T cells.

We have performed the nuclear localisation experiment in triplicate on PBMC's derived from patients and observed no obvious differences in nuclear localisation of TyrRS in endogenous conditions. These data are now also included in the manuscript (Supplementary Fig. 2b).

Page 8 line 176, (Fig. 2m): Interpretation of the data correct, however, no indication of the number of replicates and if this WB is representative of replicates. Also, quantification would greatly improve the clarity of the data. Dexamethasone seem to be less efficient than resveratrol on E2F1 acetylation, is this expected?

The experiment of Fig. 2m has been repeated three times, with independent biological samples, and quantification has been added to the new figure. Dexamethasone and resveratrol have a similar effect on reducing E2F1 acetylation.

Page 8 line 179, Supplementary Figs. 2b, 2c): number of replicates not indicated.

As indicated above, the effect of resveratrol and dexamethasone on E2F1 acetylation has now been repeated three times, with independent biological samples. The information has been added to the figure legends.

Page 9 line 205, Supplementary Table 1: Number of counts for the GOterms only accounts for 410 genes (57%) from the 700 genes that were found differentially regulated. Could the authors specify why they excluded 43% from the analysis? Alternatively, does it mean that 43% of the regulated transcriptome have less than 3 GOterm counts?

We performed a new GO analysis using fast gene set enrichment analysis (fgsea). It allows to select from an *a priori* defined list of gene sets those which have non-random behaviour in a considered experiment. The analysis will take into account the ranking of each gene (p-value and fold change difference) in order to identify the most enriched GO terms.

The table below depicts the amount of genes used throughout the analysis. The first column shows DE genes, followed by the DE genes with known gene symbols. The last column depicts the number of genes used in fgsea (having known ontologies). The main reason for the lower number of genes is that we do not limit the RNAseq analysis to the protein coding transcripts only, thus some of these transcripts do not have an associated GO term. Also, we do not purposely exclude genes from the analysis, but rather the program was unable to identify any enriched terms related to some genes, hence these genes do not show up in the final results. To clarify this issue we have adjusted the Material and Methods section in the manuscript on page 28.

	DE (padj <0.1)	DE (padj <0.1) Known gene symbol	fgsea
E_vw_w	1266	1150	474
E_vs_WT	379	338	141
WT_vs_w	961	889	352
magenta (DE E_vs_w)	151	131	62

Page 9 line 208: ...dE2F1, which was in agreement with our previous results, as well as gem... Error in the figure legend: Enriched motifs for the dE2F1 and gem transcription factors identified in the up- and down-regulated genes of the TyrRS-WT condition, respectively. SHOULD BE: Enriched motifs for the dE2F1 and gem transcription factors identified in the down- and up-regulated genes of the TyrRS-WT condition, respectively.

The original statement was correct. dE2F1 and gem transcription factors were indeed identified in the up- and down-regulated genes of the TyrRS-WT condition, respectively, as shown in Fig. 3f and Supplementary Table 2.

Page 9 line 214, Fig. 3c: As a general comment, it would be interesting to have a supp table comparing the WT and E196K GOterm enrichment that allows to have a direct readout on which network of genes sustaining a given pathway is transcriptionally down or up-regulated by the CMT TyrRS mutation.

We thank the Reviewer for this valuable suggestion. We have performed a new GO analysis using fgsea. The results for the WT vs control and the E196K vs control are summarized both in Supplementary Tables 1 and 3. Supplementary table 4 contains all the enriched GO terms (NES>1.5 and *padj*<0.01) in each condition side by side. To make a more graphical representation of this table we included Supplementary Fig. 4a. The plot shows the top 20 most enriched terms for each condition, and the subsequent alignment of these features. Red triangles are overrepresented terms. The intensity of the

colour defines the degree overrepresentation. This new figure concludes that some GO terms overlap between the mutant and the WT condition, suggesting that some biological processes could be affected as a result of a gain of function mechanism. It also reveals the GO terms that are different between both conditions, of which neuromuscular synaptic transmission is of particular interest.

Page 9 line 216, Supplementary Table 3: Number of counts for the GO terms only accounts for 330 genes (40%) from the 700 genes that were found differentially regulated. Could the authors specify why they excluded 60% from the analysis?

Similar to the question regarding Page 9 line 205, Supplementary Table 1, we have performed a new GO analysis. We also explain in the material and method section the outline of our analysis.

Page 9 line 217, Fig. 3d: Same comment as for fig 3c.

Such table and figure have been generated and are included in the new version of the manuscript.

Page 10 line 220, Supplementary Fig. 3a: Color code not explained, areas of clusters not explained.

Colour codes refer to different conditions. Area of clusters refers to confidence interval. We have now clarified this in the figure legend of Supplementary Fig. 4b.

Page 10 line 243, Fig. 4a: specify number of statistical replicates? Quantifications would have helped. Authors should indicate the concentrations of DMSO in μM rather than %.

The experiment has been repeated three times, with independent biological samples, and quantification is now added to the figure. Because DMSO does not affect TyrRS nuclear localization, we simplified the figure and removed the different DMSO lanes and restrict ourselves to one (labelled as "0" μM of Emb). We have converted the concentrations of DMSO to μM as well, which are now mentioned in the figure legends and in the material and methods section.

Page 10 line 244, Fig. 4b: How does 0-100 μM of embelin for flies compared to 0-30 μM for HEK293T cells? Also, the control (DMSO) is missing.

We do not know the exact uptake of any compound in *Drosophila* compared to cells. Furthermore, it is hard to control the amount of food each larvae or adult fly eats compared to each other. Therefore, we always perform a dose response curve first, to estimate the optimal concentrations of the drug to be used in the specific assays.

Page 11 line 248, Fig. 4c: Explain why pupae formation in absence of 0.3% DMSO for the control or for the E196K has not been measured?

We saw earlier (Figs. 2n and 2o) that there was no significant difference between flies that were fed food containing 0.3% DMSO (42 mM) or no DMSO. Therefore, we continued with the DMSO conditions as the vehicle control in all the other assays.

Page 11 line 250, Fig. 4d: Explain why there is not a dose-dependence amelioration of motor performance upon treatment with embelin, from the fig it seems that there is a 10 % amelioration

which plateauing regardless of the embelin concentration used in the experiment while there is a huge difference in nuclear import of TyrRS between 25, 50 and 100 μM , could the authors comment on this?

A similar question has been asked by both Reviewer 1 and 2 as well. We see that increasing the concentration of embelin has a dose-dependent effect on the nuclear exclusion of TyrRS. We have now performed additional experiments investigating the climbing behaviour of flies upon treatment with embelin. These additional analyses revealed a partial rescue when mutant flies were treated with 50 μM or 100 μM of embelin. Despite the continuous reduction in nuclear TyrRS, we do not observe further amelioration of the motor deficit upon further increase of embelin concentrations. Embelin has a plethora of functions such as inhibition of XIAP, resulting in anti-tumour and anti-inflammatory activity, inhibition of NF- κB , but also the inhibition of PCAF (Modak et al., 2013). Possibly, these additional functions of embelin might affect or limit the beneficial effect of the drug, i.e. can be regarded as off-target effects. Future efforts should focus on further optimising the efficacy of PCAF inhibition in order to exclude TyrRS from the nucleus without off-target effects. To this end it was important to exclude TyrRS from the nucleus using a different approach, and therefore we used the genetic elusion, which was proven to be successful.

Page 11 line 253, Supplementary Fig. 4: Authors are right but differences are not statistically significant, and there seems to be a mistake since the 2 first histograms are annotated as identical assay conditions (0 μM embelin 0.3% DMSO).

The mistake has been addressed.

Page 11 line 264, Fig. 5a: Number of replicates? True, however there is also substantially less TyrRS ΔNLS in the cytoplasm (4th lane), explain why? Also, the lane of the cytoplasmic fraction of E196K variant shows a band of higher molecular weight that is recognized by anti-TyrRS antibodies and not imported inside nuclei. Do the authors know to which form of TyrRS this band corresponds?

In general we see a lower expression of TyrRS-WT ΔNLS ; we have tested multiple lines but were never able to reach equal expression of the transgene. We have added a statement in the manuscript to clarify this. It is important to note that the expression levels of the TyrRS-E196K and TyrRS-E196K ΔNLS are similar, thus supporting our findings that nuclear exclusion of the mutant protein rescues the disease phenotypes. Despite this, we have performed the experiment three times and have quantified the amount of TyrRS in the nucleus normalized to the total amount of TyrRS. We used this method to account for the reduced band in the cytoplasm in the TyrRS-WT ΔNLS condition.

The higher MW band only appears when expressing the ΔNLS transgenes (Supplementary Fig. 6b). We speculate this is the acetylated form of TyrRS and we are happy to share our thinking on why we make this speculation. The ΔNLS mutation is made by changing the TyrRS NLS sequence from KKKLKK into NNKLNK, and the third K (underlined) was reported to be an acetylation site, modification of which is critical for regulating TyrRS nuclear import (Cao et al., 2017). Mutating the neighbour Ks to Ns probably makes the third K more accessible for acetylation, hence the appearance of the slower running band in the ΔNLS transgenes. Although the acetylation per se should enhance TyrRS nuclear import (Cao et al., 2017), mutating the other Ks to Ns disrupted the NLS. Therefore, ΔNLS (NNKLNK) results in a TyrRS

protein with an increased acetylation level, however, with decreased nuclear localization. We now added this speculation into the figure legend of Supplementary Fig. 6b.

Page 12 line 282, Fig. 5d-f: In fig 5e, are the NMJs data between WT E196K significant? This would be needed to sustain the authors' conclusions.

We have made a more precise and accurate measurement of the NMJ size. In the original manuscript, we extracted the NMJ size as a projected area from the coordinates of the boutons using the smallest bounding rectangle method. While this is a fast, correct and reliable method, it is influenced by the NMJ shape. A more precise extraction of the 'NMJ size' is the length of the NMJ axons projecting on the muscle. To measure this, we re-evaluated the available images of the NMJ's and measured the total length of the axons in each NMJ by manually tracing the axons based on the HRP channel (green signal in Fig. 5d). It is now more evident from the updated plots of Figs. 5e, 5f that there is a clear difference between the TyrRS-196K condition and the controls. We have also adjusted the material and methods section regarding this new analysis.

References

- Atkinson, D., et al., 2017. Sphingosine 1-phosphate lyase deficiency causes Charcot-Marie-Tooth neuropathy. *Neurology*. 88, 533-542.
- Cao, X., et al., 2017. Acetylation promotes TyrRS nuclear translocation to prevent oxidative damage. *Proc Natl Acad Sci U S A*. 114, 687-692.
- Castells-Nobau, A., et al., 2017. Two Algorithms for High-throughput and Multi-parametric Quantification of Drosophila Neuromuscular Junction Morphology. *J Vis Exp*.
- Chang, S., et al., 2008. Identification of small molecules rescuing fragile X syndrome phenotypes in Drosophila. *Nat Chem Biol*. 4, 256-63.
- Ermanoska, B., et al., 2014. CMT-associated mutations in glycyl- and tyrosyl-tRNA synthetases exhibit similar pattern of toxicity and share common genetic modifiers in Drosophila. *Neurobiol Dis*. 68, 180-9.
- Froelich, C.A., First, E.A., 2011. Dominant Intermediate Charcot-Marie-Tooth disorder is not due to a catalytic defect in tyrosyl-tRNA synthetase. *Biochemistry*. 50, 7132-45.
- Goedhart, J., 2019. PlotsOfDifferences – a web app for the quantitative comparison of unpaired data. *bioRxiv*.
- Jia, H., et al., 2008. High doses of nicotinamide prevent oxidative mitochondrial dysfunction in a cellular model and improve motor deficit in a Drosophila model of Parkinson's disease. *J Neurosci Res*. 86, 2083-90.
- Leitao-Goncalves, R., et al., 2012. Drosophila as a platform to predict the pathogenicity of novel aminoacyl-tRNA synthetase mutations in CMT. *Amino Acids*. 42, 1661-8.
- Modak, R., et al., 2013. Probing p300/CBP associated factor (PCAF)-dependent pathways with a small molecule inhibitor. *ACS Chem Biol*. 8, 1311-23.
- Qurashi, A., et al., 2012. Chemical screen reveals small molecules suppressing fragile X premutation rCGG repeat-mediated neurodegeneration in Drosophila. *Hum Mol Genet*. 21, 2068-75.
- Storkebaum, E., et al., 2009. Dominant mutations in the tyrosyl-tRNA synthetase gene recapitulate in Drosophila features of human Charcot-Marie-Tooth neuropathy. *Proc Natl Acad Sci U S A*. 106, 11782-7.
- Wei, N., et al., 2014. Oxidative stress diverts tRNA synthetase to nucleus for protection against DNA damage. *Mol Cell*. 56, 323-32.

REVIEWERS' COMMENTS:

Reviewer #1 (Remarks to the Author):

The authors have responded to the critiques to the best of their ability. Data presentation and reliability issues have been addressed, and these changes improve the manuscript. The in vivo effect sizes remain modest (in fairness to the authors the synaptic changes are similar to ones published as significant in the field) and in some cases (Figure 1a-l) unconvincing. The issue of the degree of advance the work represents from prior work from the laboratory remains unchanged.

Reviewer #2 (Remarks to the Author):

The authors have done an excellent and thorough job addressing the initial reviewer comments including new statistical analyses, new experiments, and improved data interpretation.

Reviewer #3 (Remarks to the Author):

In their revised version that was submitted, Bervoets and coworkers answered in a satisfactory manner to all the comments I made and the new data on the morphology and electrophysiological measurements of the synaptic transmission of the giant fibres, strengthened the author's conclusions that the CMT phenotypes they studied originate from the nuclear-translocated mutant TyRSs. I, therefore, recommend publication of this revised manuscript as it is in Nature Communications.

Typo: Line 317, "this" has to be removed

Point-by-point Response to Reviewers' comments:

Reviewer #1:

The authors have responded to the critiques to the best of their ability. Data presentation and reliability issues have been addressed, and these changes improve the manuscript. The in vivo effect sizes remain modest (in fairness to the authors the synaptic changes are similar to ones published as significant in the field) and in some cases (Figure 1a-l) unconvincing. The issue of the degree of advance the work represents from prior work from the laboratory remains unchanged.

We appreciate the comments of the reviewer and the acknowledgment that we have improved the manuscript.

The comment that “The in vivo effect sizes remain...in some cases (Figure 1a-l) unconvincing” likely refers to Figure 2a-l. We have thoroughly responded to this comment in the last rebuttal. The key points were – 1) using eye surface area as the parameter we can clearly demonstrate a genetic interaction between E2F1/Dp and TyrRS-E196K, because neither the expression of E2F1/Dp nor the expression of TyrRS-E196K causes a change in eye size, however, co-overexpression of E2F1/Dp and TyrRS-E196K significantly reduces the eye size. 2) The genetic interaction study presented on Figure 2a-l is only one aspect of demonstrating the interaction between the E2F1 transcriptional complex and mutant TyrRS. We have presented additional evidences for such interaction using alternative approaches throughout the manuscript.

The last point of the reviewer questions whether the paper represents a significant advancement beyond our prior work. We would like to emphasize that our previous work only revealed that wild-type TyrRS has a nuclear function in protecting cells against DNA damage, but did not connect the nuclear function of TyrRS to any disease. This work not only linked the nuclear TyrRS to Charcot-Marie-Tooth disease, but also demonstrated that the nuclear function of TyrRS is beyond the previously identified DNA damage protection function. Importantly, this is the first report demonstrating human disease association of nucleus-localized tRNA synthetase. Therefore, this work is of general interest for the fields of translation and transcription and for the basic and translational aspects of molecular neurodegeneration.

Reviewer #2:

The authors have done an excellent and thorough job addressing the initial reviewer comments including new statistical analyses, new experiments, and improved data interpretation.

We appreciate the comments of the reviewer.

Reviewer #3 (Remarks to the Author):

In their revised version that was submitted, Bervoets and coworkers answered in a satisfactory manner

to all the comments I made and the new data on the morphology and electrophysiological measurements of the synaptic transmission of the giant fibres, strengthened the author's conclusions that the CMT phenotypes they studied originate from the nuclear-translocated mutant TyRSs. I, therefore, recommend publication of this revised manuscript as it is in Nature Communications.
Typo: Line 317, "this" has to be removed

We appreciate the comments of the reviewer. The typo is corrected in the revised manuscript.